# Improved off-policy training of diffusion samplers

**Marcin Sendera**
Mila, Université de Montréal
Jagiellonian University

**Minsu Kim**
Mila, Université de Montréal
KAIST

**Sarthak Mittal**
Mila, Université de Montréal

**Pablo Lemos**
Mila, Université de Montréal
Ciela Institute
Dreamfold

**Luca Scimeca**
Mila, Université de Montréal

**Jarrid Rector-Brooks**
Mila, Université de Montréal
Dreamfold

**Alexandre Adam**
Mila, Université de Montréal
Ciela Institute

**Yoshua Bengio**
Mila, Université de Montréal
CIFAR

**Nikolay Malkin**
Mila, Université de Montréal
University of Edinburgh

{marcin.sendera,...,nikolay.malkin}@mila.quebec

## Abstract

We study the problem of training diffusion models to sample from a distribution with a given unnormalized density or energy function. We benchmark several diffusion-structured inference methods, including simulation-based variational approaches and off-policy methods (continuous generative flow networks). Our results shed light on the relative advantages of existing algorithms while bringing into question some claims from past work. We also propose a novel exploration strategy for off-policy methods, based on local search in the target space with the use of a replay buffer, and show that it improves the quality of samples on a variety of target distributions. Our code for the sampling methods and benchmarks studied is made public at (link) as a base for future work on diffusion models for amortized inference.

## 1 Introduction

Approximating and sampling from complex multivariate distributions is a fundamental problem in probabilistic deep learning [*e.g.*, 27, 35, 26, 48, 57] and in scientific applications [3, 52, 38, 1, 32]. The problem of drawing samples from a distribution given only an unnormalized probability density or energy is particularly challenging in high-dimensional spaces and when the distribution of interest has many separated modes [5]. Sampling methods based on Markov chain Monte Carlo (MCMC) – such as Metropolis-adjusted Langevin [MALA; 24, 65, 64] and Hamiltonian MC [HMC; 20, 31] – may be slow to mix between modes and have a high cost per sample. While variants such as sequential MC [SMC; 25, 13, 16] and nested sampling [69, 10, 43] have better mode coverage, their cost may grow prohibitively with the dimensionality of the problem. This motivates the use of amortized variational inference, *i.e.*, fitting parametric models that sample the target distribution.

Diffusion models, continuous-time stochastic processes that gradually evolve a simple distribution to a complex target, are powerful density estimators with proven mode-mixing properties [15]; as such, they have been widely used in the setting of generative models learned from data [70, 72, 28, 50, 66]. However, the problem of training diffusion models to sample from a distribution with a given black-box density or energy function has attracted less attention. Recent work has drawn connections between diffusion (learning the denoising process) and stochastic control (learning the Föllmer

drift [21]), leading to approaches such as the path integral sampler [PIS; 88], denoising diffusion sampler [DDS; 78], and time-reversed diffusion sampler [DIS; 8]; such approaches were recently unified by [63] and [79]. Another line of work [42, 86] is based on continuous generative flow networks (GFlowNets), which are deep reinforcement learning algorithms adapted to variational inference that offer stable off-policy training and thus flexible exploration [46].

Despite the advances in sampling methods and attempts to unify them theoretically [63, 79], the field suffers from some failures in benchmarking and reproducibility, with the works differing in the choice of model architectures, using unstated hyperparameters, and even disagreeing in their definitions of the same target densities (see §B.1). The **first main contribution** of this paper is a unified library for diffusion-structured samplers. The library has a focus on off-policy methods (continuous GFlowNets) but also includes simulation-based variational objectives such as PIS. Using this codebase, we are able to benchmark methods from past work under comparable conditions and confirm claims about exploration strategies and desirable inductive biases, while calling into question other claims on robustness and sample efficiency. Our library also includes several new modeling and training techniques, and we provide preliminary evidence of their utility in possible future work (§5.3).

Our **second contribution** is a study of methods for improving exploration and credit assignment – the propagation of learning signals from the target density to the parameters of earlier sampling steps – in diffusion-structured samplers (§4). First, our results (§5.2) suggest that the technique of utilizing partial trajectory information [44, 55], as done in the diffusion setting by [86], offers little benefit, and a higher training cost, over on-policy [88] or off-policy [42] trajectory-based optimization. Second, we examine the utility of a gradient-based variant which parametrizes the denoising distribution as a correction to a Langevin process [88]. We show that this inductive bias is also beneficial in the off-policy (GFlowNet) setting despite higher computational cost. Finally, motivated by recent approaches in discrete sampling, we propose an efficient exploration technique based on local search in the target space with the use of a replay buffer, which improves sample quality across various target distributions.

## 2   Prior work

**Amortized variational inference** approaches use a parametric model $q_\theta$ to approximate a given target density $p_{target}$, typically through stochastic optimization [30, 58, 2]. Notably, explicit density models like autoregressive models and normalizing flows have been extensively utilized in density estimation [60, 19, 81, 22, 51]. However, these models impose structural constraints, thereby limiting their expressive power [14, 23, 87]. The adoption of **diffusion processes** in generative models has stimulated a renewed interest in hierarchical models as density estimators [80, 28, 76]. Approaches like PIS [88] leverage stochastic optimal control for sampling from unnormalized densities, albeit still struggling with scalability in high-dimensional spaces.

**Generative flow networks**, originally defined in the discrete case by [6, 7], view hierarchical sampling (*i.e.*, stepwise generation) as a sequential decision-making process and represent a synthesis of reinforcement learning and variational inference approaches [46, 90, 73, 18], expanding from specific scientific domains [*e.g.*, 36, 4, 89] to amortized inference over a broader array of latent structures [*e.g.*, 77, 34]. Their ability to efficiently navigate trajectory spaces via off-policy exploration has been crucial, yet they encounter challenges in training dynamics, such as credit assignment and exploration efficiency [45, 44, 55, 59, 68, 39, 37]. These challenges have repercussions in the scalability of these methods in more complex scenarios, which this paper addresses in the continuous case.

## 3   Setting: Diffusion-structured sampling

Let $\mathcal{E} : \mathbb{R}^d \to \mathbb{R}$ be a differentiable *energy function* and define $R(\mathbf{x}) = \exp(-\mathcal{E}(\mathbf{x}))$, the *reward* or *unnormalized target density*. Assuming the integral $Z := \int_{\mathbb{R}^d} R(\mathbf{x}) \, d\mathbf{x}$ exists, $\mathcal{E}$ defines a *Boltzmann density* $p_{target}(\mathbf{x}) = R(\mathbf{x})/Z$ on $\mathbb{R}^d$. We are interested in the problems of sampling from $p_{target}$ and approximating the *partition function* $Z$ given access only to $\mathcal{E}$ and possibly to its gradient $\nabla \mathcal{E}$.

We describe two closely related perspectives on this problem: via neural SDEs and stochastic control (§3.1) and via continuous generative flow networks (§3.2).

## 3.1 Euler-Maruyama hierarchical samplers

**Generative modeling with SDEs.** Diffusion models assume a continuous-time generative process given by a neural stochastic differential equation [SDE; 75, 54, 67]:

$$d\mathbf{x}_t = u(\mathbf{x}_t, t; \theta) \, dt + g(\mathbf{x}_t, t; \theta) \, d\mathbf{w}_t, \tag{1}$$

where $\mathbf{x}_0$ follows a fixed tractable distribution $\mu_0$ (such as a Gaussian or a point mass). The initial distribution $\mu_0$ and the stochastic dynamics specified by (1) induce marginal densities $p_t$ on $\mathbb{R}^d$ for each $t > 0$. The functions $u$ and $g$ have learnable parameters that we wish to optimize, using some objective, so as to make the terminal density $p_1$ close to $p_{\text{target}}$. Samples can be drawn from $p_1$ by sampling $\mathbf{x}_0 \sim \mu_0$ and simulating the SDE (1) to time $t = 1$.

The SDE driving $\mu_0$ to $p_{\text{target}}$ is not unique. However, if one fixes a reverse-time SDE, or *noising process*, that pushes $p_{\text{target}}$ at $t = 1$ to $\mu_0$ at $t = 0$, then its reverse, the forward SDE (1), is uniquely determined under mild conditions and is called the *denoising process*. For usual choices of the noising process, there are stochastic regression objectives for learning the drift $u$ of the denoising process *given samples from $p_{\text{target}}$*, and the diffusion rate $g$ is available in closed form [28, 72].

**Time discretization.** In practice, the integration of the SDE (1) is approximated by a discrete-time scheme, the simplest of which is Euler-Maruyama integration. The process (1) is replaced by a discrete-time Markov chain $\mathbf{x}_0 \to \mathbf{x}_{\Delta t} \to \mathbf{x}_{2\Delta t} \to \cdots \to \mathbf{x}_1$, where $\Delta t = \frac{1}{T}$ is the time increment and and $T$ is the number of steps:

$$\mathbf{x}_0 \sim \mu_0, \quad \mathbf{x}_{t+\Delta t} = \mathbf{x}_t + u(\mathbf{x}_t, t; \theta)\Delta t + g(\mathbf{x}_t, t; \theta)\sqrt{\Delta t}\, \mathbf{z}_t \quad \mathbf{z}_t \sim \mathcal{N}(\mathbf{0}, \mathbf{I}_d). \tag{2}$$

The density of the transition kernel from $\mathbf{x}_t$ to $\mathbf{x}_{t+\Delta t}$ can explicitly be written as

$$p_F(\mathbf{x}_{t+\Delta t} \mid \mathbf{x}_t) = \mathcal{N}(\mathbf{x}_{t+\Delta t}; \mathbf{x}_t + u(\mathbf{x}_t, t; \theta)\Delta t, g(\mathbf{x}_t, t; \theta)^2 \Delta t \mathbf{I}_d), \tag{3}$$

where $p_F$ denotes the transition density of the discretized forward SDE. This density defines a joint distribution over trajectories starting at $\mathbf{x}_0$:

$$p_F(\mathbf{x}_{\Delta t}, \ldots, \mathbf{x}_1 \mid \mathbf{x}_0) = \prod_{i=0}^{T-1} p_F(\mathbf{x}_{(i+1)\Delta t} \mid \mathbf{x}_{i\Delta t}). \tag{4}$$

Similarly, a discrete-time reverse process $\mathbf{x}_1 \to \mathbf{x}_{1-\Delta t} \to \mathbf{x}_{1-2\Delta t} \to \cdots \to \mathbf{x}_0$ with transition densities $p_B(\mathbf{x}_{t-\Delta t} \mid \mathbf{x}_t)$ defines a joint distribution[1] via

$$p_B(\mathbf{x}_0, \ldots, \mathbf{x}_{1-\Delta t} \mid \mathbf{x}_1) = \prod_{t=1}^{T} p_B(\mathbf{x}_{(i-1)\Delta t} \mid \mathbf{x}_{i\Delta t}). \tag{5}$$

If the forward and backward processes (starting from $\mu_0$ and $p_{\text{target}}$, respectively) are reverses of each other, then they define the same distribution over trajectories, *i.e.*, for all $\mathbf{x}_0 \to \mathbf{x}_{\Delta t} \to \cdots \to \mathbf{x}_1$,

$$\mu_0(\mathbf{x}_0)p_F(\mathbf{x}_{\Delta t}, \ldots, \mathbf{x}_1 \mid \mathbf{x}_0) = p_{\text{target}}(\mathbf{x}_1)p_B(\mathbf{x}_0, \ldots, \mathbf{x}_{1-\Delta t} \mid \mathbf{x}_1). \tag{6}$$

In particular, the marginal densities of $\mathbf{x}_1$ under the forward and backward processes are then equal to $p_{\text{target}}$, and the forward process can be used to sample the target distribution.

Because the reverse of a process with Gaussian increments is, in general, not itself Gaussian, (6) can be enforced only approximately, but the discrepancy vanishes as $\Delta t \to 0$ (*i.e.*, increments are infinitesimally Gaussian), an application of the central limit theorem that is key to stochastic calculus [54].

**SDE learning as hierarchical variational inference.** The problem of learning the parameters $\theta$ of the forward process so as to enforce (6) is one of hierarchical variational inference. The backward process transforms $\mathbf{x}_1$ into $\mathbf{x}_0$ via a sequence of latent variables $\mathbf{x}_{1-\Delta t}, \ldots, \mathbf{x}_0$, and the forward process aims to match the posterior distribution over these variables and thus to approximately enforce (6).

In the setting of diffusion models learned from data, where one has samples from $p_{\text{target}}$, one can optimize the forward process by minimizing the KL divergence $D_{\text{KL}}(p_{\text{target}} \cdot p_B \| \mu_0 \cdot p_F)$ between the distribution over trajectories given by the reverse process and that given by the forward process.

---

[1]In the case that $\mu_0$ is a point mass, we assume the distribution $\mathbf{x}_0 \mid \mathbf{x}_{\Delta t}$ to also be a point mass, which has density $p_B(\mathbf{x}_0 \mid \mathbf{x}_{\Delta t}) = 1$ with respect to the measure $\mu_0$.

This is equivalent to the typical training of diffusion models, which optimizes a variational bound on the data log-likelihood (see [71]). However, in the setting of an intractable density $p_{\text{target}}$, unbiased estimators of this divergence are not available. Instead, one can optimize the reverse KL:[2]

$$D_{\text{KL}}(\mu_0 \cdot p_F \| p_{\text{target}} \cdot p_B)$$

$$= \int \log \frac{\mu_0(\mathbf{x}_0) p_F(\mathbf{x}_{\Delta t}, \dots, \mathbf{x}_1 \mid \mathbf{x}_0)}{p_{\text{target}}(\mathbf{x}_1) p_B(\mathbf{x}_0, \dots, \mathbf{x}_{1-\Delta t} \mid \mathbf{x}_1)} d\mu_0(\mathbf{x}_0) p_F(\mathbf{x}_{\Delta t}, \dots, \mathbf{x}_1 \mid \mathbf{x}_0) \ d\mathbf{x}_{\Delta t} \ \dots \ d\mathbf{x}_1. \quad (7)$$

Various estimators of this objective are available. For instance, the path integral sampler objective [PIS; 88] uses the reparametrization trick to express (7) as an expectation over noise variables $\mathbf{z}_t$ that participate in the hierarchical sampling of $\mathbf{x}_{\Delta t}, \dots, \mathbf{x}_1$, yielding an unbiased gradient estimator, but one that requires backpropagation into the simulation of the forward process. The related denoising diffusion sampler [DDS; 78] applies the same principle in a different integration scheme.

## 3.2 Euler-Maruyama samplers as GFlowNets

Continuous generative flow networks (GFlowNets) [42] express the problem of enforcing (6) as a reinforcement learning task. In this section, we summarize this interpretation, its connection to neural SDEs, the associated learning objectives, and their relative advantages and disadvantages.

The connection between generative flow networks and diffusion models or SDEs was first made informally by [46] in the distribution-matching setting and by [84] in the maximum-likelihood setting, while the theoretical foundations for continuous GFlowNets were later laid down by [42].

**State and action space.** To formulate sampling as a sequential decision-making problem, one must define the spaces of states and actions. In the case of sampling by $T$-step Euler-Maruyama integration, assuming $\mu_0$ is a point mass at $\mathbf{0}$, the state space is

$$\mathcal{S} = \{(\mathbf{0}, 0) \cup \{(\mathbf{x}, t) : \mathbf{x} \in \mathbb{R}^d, t \in \{\Delta t, 2\Delta t, \dots, 1\}\}\,,$$

with the point $(\mathbf{x}, t)$ representing that the sampling agent is at position $\mathbf{x}$ at time $t$.

Sampling begins with the *initial state* $\mathbf{x}_0 := (\mathbf{0}, 0)$, proceeds through a sequence of states $(\mathbf{x}_{\Delta t}, \Delta t)$, $(\mathbf{x}_{2\Delta t}, 2\Delta t), \dots$, and ends at a state $(\mathbf{x}_1, 1)$; states $(\mathbf{x}, t)$ with $t = 1$ are called *terminal states* and their collection is denoted $\mathcal{X}$. From now on, we will often write $\mathbf{x}_t$ in place of the state $(\mathbf{x}_t, t)$ when the time $t$ is clear from context. The sequence of states $\mathbf{x}_0 \rightarrow \mathbf{x}_{\Delta t} \rightarrow \cdots \rightarrow \mathbf{x}_1$ is called a *complete trajectory*.

The *actions* from a nonterminal state $(\mathbf{x}_t, t)$ correspond to the possible next states $(\mathbf{x}_{t+\Delta t}, t + \Delta t)$ that can be reached from $(\mathbf{x}_t, t)$ by a single step of the Euler-Maruyama integrator.[3]

**Forward policy and learning problem.** A *(forward) policy* is a collection of continuous distributions over the successor states – states reachable by a single action – of every nonterminal state $(\mathbf{x}, t)$. In our context, this amounts to a collection of conditional probability densities $p_F(\mathbf{x}_{t+\Delta t} \mid \mathbf{x}_t; \theta)$, representing the density of the transition kernel from $\mathbf{x}_t$ to $\mathbf{x}_{t+\Delta t}$. GFlowNet training optimizes the parameters $\theta$, which may be the weights of a neural network specifying a density over $\mathbf{x}_{t+\Delta t}$ conditioned on $\mathbf{x}_{\Delta t}$.

A policy $p_F$ induces a distribution over complete trajectories $\tau = (\mathbf{x}_0 \rightarrow \mathbf{x}_{\Delta t} \rightarrow \cdots \rightarrow \mathbf{x}_1)$ via

$$p_F(\tau; \theta) = \prod_{i=0}^{T-1} p_F(\mathbf{x}_{(i+1)\Delta t} \mid \mathbf{x}_{i\Delta t}; \theta).$$

In particular, we get a marginal density over terminal states:

$$p_F^\top(\mathbf{x}_1; \theta) = \int p_F(\mathbf{x}_0 \rightarrow \mathbf{x}_{\Delta t} \rightarrow \cdots \rightarrow \mathbf{x}_1; \theta) \, d\mathbf{x}_{\Delta t} \dots d\mathbf{x}_{1-\Delta t}. \quad (8)$$

The learning problem solved by GFlowNets is to find the parameters $\theta$ of a policy $p_F$ whose terminating density $p_F^\top$ is equal to $p_{\text{target}}$, *i.e.*,

$$p_F^\top(\mathbf{x}_1; \theta) = \frac{R(\mathbf{x}_1)}{Z} \quad \forall \mathbf{x}_1 \in \mathbb{R}^d. \quad (9)$$

---

[2]To be precise, the fraction in (7) should be understood as a Radon-Nikodym derivative, which makes sense whether $\mu_0$ is a point mass or a continuous distribution and generalizes to continuous time [8, 63].

[3]Formally, the foundations in [42] require assuming *reference measures* with respect to which the reward and kernel densities are defined. As we deal with Euclidean spaces and assume the Lebesgue measure, readers need not burden themselves with measure theory. We note, however, that this flexibility allows easy generalization to sampling on other spaces, such as any Riemannian manifolds, where other methods do not directly apply.

However, because the integral (8) is intractable and $Z$ is unknown, auxiliary objects must be introduced into optimization objectives to enforce (9), as discussed below.

Notably, if the policy is a Gaussian with mean and variance given by neural networks taking $\mathbf{x}_t$ and $t$ as input, then learning the policy amounts to learning the drift $u(\mathbf{x}_t, t; \theta)$ and diffusion $g(\mathbf{x}_t, t; \theta)$ of a SDE (1), *i.e.*, fitting a neural SDE. **The SDE learning problem in §3.1 is thus the same as that of fitting a GFlowNet with Gaussian policies.**

**Backward policy and trajectory balance.** A *backward policy* is a collection of conditional probability densities $p_B(\mathbf{x}_{t-\Delta t} \mid \mathbf{x}_t; \psi)$, representing a probability density of transitioning from $\mathbf{x}_t$ to an ancestor state $\mathbf{x}_{t-\Delta t}$. The backward policy induces a distribution over complete trajectories $\tau$ conditioned on their terminal state (cf. (5)):

$$p_B(\tau \mid \mathbf{x}_1; \psi) = \prod_{i=1}^{T} p_B(\mathbf{x}_{(i-1)\Delta t} \mid \mathbf{x}_{i\Delta t}; \psi),$$

where exceptionally $p_B(\mathbf{x}_0 \mid \mathbf{x}_{\Delta t}) = 1$ as $\mu_0$ is a point mass.

Generalizing a result in the discrete-space setting [45], [42] show that $p_F$ samples from the target distribution (*i.e.*, satisfies (9)) if and only if there exists a backward policy $p_B$ and a scalar $Z_\theta$ such that the *trajectory balance conditions* are fulfilled for every complete trajectory $\tau = (\mathbf{x}_0 \to \mathbf{x}_{\Delta t} \to \cdots \to \mathbf{x}_1)$:

$$Z_\theta p_F(\tau; \theta) = R(\mathbf{x}_1) p_B(\tau \mid \mathbf{x}_1; \psi). \tag{10}$$

If these conditions hold, then $Z_\theta$ equals the true partition function $Z = \int_{\mathbf{x}} R(\mathbf{x}) \, d\mathbf{x}$. The *trajectory balance objective* for a trajectory $\tau$ is the squared log-ratio of the two sides of (10), that is:

$$\mathcal{L}_{\text{TB}}(\tau; \theta, \psi) = \left( \log \frac{Z_\theta p_F(\tau; \theta)}{R(\mathbf{x}_1) p_B(\tau \mid \mathbf{x}_1; \psi)} \right)^2. \tag{11}$$

One can thus achieve (9) by minimizing to zero the loss $\mathcal{L}_{\text{TB}}(\tau; \theta, \psi)$ with respect to the parameters $\theta$ and $\psi$, where the trajectories $\tau$ used for training are sampled from some *training policy* $\pi(\tau)$. While it is possible to optimize (11) with respect to the parameters of both the forward and backward policies, in some learning problems, one *fixes* the backward policy and only optimizes the parameters of $p_F$ and the estimate of the partition function $Z_\theta$. For example, for most experiments in §5, we fix the backward policy to a discretized Brownian bridge, following past work.

**Off-policy optimization.** Unlike the KL objective (7), whose gradient involves an expectation over the distribution of trajectories under the current forward process, (11) can be optimized off-policy, *i.e.*, using trajectories sampled from an arbitrary distribution $\pi$. Because minimizing $\mathcal{L}_{\text{TB}}(\tau; \theta, \psi)$ to 0 for all $\tau$ in the support of $\pi$ will achieve (9), $\pi$ can be taken be any distribution with full support, so as to promote discovery of modes of the target distribution. Various choices motivated by reinforcement learning techniques have been proposed, including noisy exploration or tempering [6], replay buffers [17], Thompson sampling [59], and backward traces from terminal states obtained by MCMC [43]. In the continuous case, [46, 42] proposed to simply add a small constant to the policy variance when sampling trajectories for training. Off-policy optimization is a key advantage of GFlowNets over variational methods such as PIS, which require on-policy optimization [46].

However, when $\mathcal{L}_{\text{TB}}$ happens to be optimized on-policy, *i.e.*, using trajectories sampled from the policy $p_F$ itself, we get an unbiased estimator of the gradient of the KL divergence (7) with respect to $p_F$'s parameters up to a constant [62, 46, 90], that is:

$$\mathbb{E}_{\tau \sim p_F(\tau)} \left[ \nabla_{\theta'} \mathcal{L}_{\text{TB}}(\tau; \theta, \psi) \right] = 2 \nabla_{\theta'} D_{\text{KL}}(p_F(\tau; \theta) \| p_{\text{target}}(\mathbf{x}_1) p_B(\tau \mid \mathbf{x}_1; \psi)),$$

where $\nabla_{\theta'}$ denotes the gradient with respect to the parameters of $p_F$, but not $Z_\theta$. This unbiased estimator tends to have higher variance than the reparametrization-based estimator used by PIS. On the other hand, it does not require backpropagation through the simulation of the forward process and can be used to optimize the parameters of both the forward and backward policies.

**Other objectives.** The trajectory balance objective (11) is not the only possible objective that can be used to enforce (9). A notable generalization is *subtrajectory balance* [SubTB; 44], which involves modeling a scalar *state flow* $f(\mathbf{x}_t; \theta)$ associated with each state $\mathbf{x}_t$ – intended to model the marginal density of the forward process at $\mathbf{x}_t$ – and enforcing *subtrajectory balance* conditions for all partial trajectories $\mathbf{x}_{m\Delta t} \to \mathbf{x}_{(m+1)\Delta t} \to \cdots \to \mathbf{x}_{n\Delta t}$:

$$f(\mathbf{x}_{m\Delta t}; \theta) \prod_{i=m}^{n-1} p_F(\mathbf{x}_{(i+1)\Delta t} \mid \mathbf{x}_{i\Delta t}; \theta) = f(\mathbf{x}_{n\Delta t}; \theta) \prod_{i=m+1}^{n} p_B(\mathbf{x}_{(i-1)\Delta t} \mid \mathbf{x}_{i\Delta t}; \psi), \tag{12}$$

where for terminal states $f(\mathbf{x}_1) = R(\mathbf{x}_1)$. This approach has some computational overhead associated with training the state flow, but has been shown to be effective in discrete-space settings, especially when combined with the *forward-looking* reward shaping scheme proposed by [55]. It has also been tested in the continuous case, but our experimental results suggest that it offers little benefit over the TB objective in the diffusion setting (see §4.1 and §B.1).

It is also worth noting the off-policy VarGrad estimator [53, 62], rediscovered for GFlowNets by [85]. Like TB, VarGrad can be optimized over trajectories drawn off-policy. Rather than enforcing (10) for every trajectory, VarGrad optimizes the empirical *variance* (over a minibatch) of the log-ratio of the two sides of (10). As noted by [46], this is equivalent to minimizing $\mathcal{L}_{\text{TB}}$ first with respect to $\log Z_\theta$ to optimality over the batch, then with respect to the parameters of $p_F$.

# 4 Exploration and credit assignment in continuous GFlowNets

The main challenges in training off-policy sampling models are exploration efficiency (discovery of high-reward states) and credit assignment (propagation of reward signals to the actions that led to them). We describe several new and existing methods for addressing these challenges in the context of diffusion-structured GFlowNets. These techniques will be empirically studied and compared in §5.

## 4.1 Credit assignment methods

**Partial energies and subtrajectory-based learning.** [86] studied the diffusion sampler learning problem introduced by [42], but replaced the TB learning objective with the SubTB objective.[4] In addition, an inductive bias resembling the geometric interpolation in [47] was used for the state flow function:

$$\log f(\mathbf{x}_t; \theta) = (1 - t) \log p_t^{\text{ref}}(\mathbf{x}_t) + t \log R(\mathbf{x}_t) + \text{NN}(\mathbf{x}_t, t; \theta), \tag{13}$$

where NN is a neural network and $p_t^{\text{ref}}(\mathbf{x}_t) = \mathcal{N}(\mathbf{x}_t; 0, \sigma^2 t I_d)$ is the marginal density of a Brownian motion with rate $\sigma$ at $\mathbf{x}_t$. The use of the target density $\log R(\mathbf{x}_t) = -\mathcal{E}(\mathbf{x}_t)$ in the state flow function was hypothesized to provide an effective signal driving the sampler to high-density states at early steps in the trajectory. Such an inductive bias on the state flow was called *forward-looking* (FL) by [55], and we will refer to this method as **FL-SubTB** in §5.

**Langevin dynamics inductive bias.** [88] proposed an inductive bias on the architecture of the drift of the neural SDE $u(\mathbf{x}_t, t; \theta)$ (in GFlowNet terms, the mean of the Gaussian density $p_F(\mathbf{x}_{t+\Delta t} \mid \mathbf{x}_t; \theta)$) that resembles a Langevin process on the target distribution. One writes

$$u(\mathbf{x}_t, t; \theta) = \text{NN}_1(\mathbf{x}_t, t; \theta) + \text{NN}_2(t; \theta)\nabla\mathcal{E}(\mathbf{x}_t), \tag{14}$$

where $\text{NN}_1$ and $\text{NN}_2$ are neural networks outputting a vector and a scalar, respectively. The second term in (14) is a scaled gradient of the target energy – the drift of a Langevin SDE – and the first term is a learned correction. This inductive bias, which we name the *Langevin parametrization* (**LP**), was shown to improve the efficiency of PIS. We will study its effect on continuous GFlowNets in §5.

The inductive bias (14) placed on policies represents a different way of incorporating the reward signal at intermediate steps in the trajectory and can steer the sampler towards low-energy regions. It contrasts with (13) in that it provides the gradient of the energy directly to the policy, rather than just using the energy to provide a learning signal to policies via the parametrization of the log-state flow (13).

Considerations of the continuous-time limit lead us to conjecture that the Langevin parametrization (14) with $\text{NN}_1$ independent of $\mathbf{x}_t$ is *equivalent* to the forward-looking flow (13) in the limit of small time increments $\Delta t \to 0$, *i.e.*, they induce the same asymptotics of the discrepancy in the SubTB constraints (12) over short partial trajectories. Such theoretical analysis can be the subject of future work.

## 4.2 A new method for off-policy exploration with local search and replay buffer

**Local search with parallel MALA.** The FL and LP inductive biases both induce computational overhead: either in the evaluation and optimization of a state flow or in the need to evaluate the energy gradient at every step of sampling (see §C.3). We present an alternative technique that does not induce additional computation cost per training trajectory.

---

[4]Despite the claimed benefits of FL-SubTB for diffusion samplers, we discovered that [86] modifies critical experimental variables in comparisons and reports irreproducible results; see §B.1.

Table 1: Log-partition function estimation errors for unconditional modeling tasks (mean and standard deviation over 5 runs). The four groups of models are: MCMC-based samplers, simulation-driven variational methods, baseline GFlowNet methods with different learning objectives, and methods augmented with Langevin parametrization and local search. See §C.1 for additional metrics.

| Energy → | 25GMM ($d=2$) | | Funnel ($d=10$) | | Manywell ($d=32$) | | LGCP ($d=1600$) | |
|---|---|---|---|---|---|---|---|---|
| Algorithm ↓ Metric → | $\Delta \log Z$ | $\Delta \log Z^{\mathrm{RW}}$ | $\Delta \log Z$ | $\Delta \log Z^{\mathrm{RW}}$ | $\Delta \log Z$ | $\Delta \log Z^{\mathrm{RW}}$ | $\log \hat{Z}$ | $\log \hat{Z}^{\mathrm{RW}}$ |
| SMC | $0.569_{\pm 0.010}$ | | $0.561_{\pm 0.801}$ | | $14.99_{\pm 1.078}$ | | *See discussion in §B.1* | |
| GGNS [43] | $0.016_{\pm 0.042}$ | | $0.033_{\pm 0.173}$ | | $0.292_{\pm 0.454}$ | | N/A | |
| DIS [8] | $1.125_{\pm 0.056}$ | $0.986_{\pm 0.011}$ | $0.839_{\pm 0.169}$ | $0.093_{\pm 0.038}$ | $10.52_{\pm 1.02}$ | $3.05_{\pm 0.46}$ | $299.83_{\pm 0.67}$ | $361.15_{\pm 6.48}$ |
| DDS [78] | $1.760_{\pm 0.08}$ | $0.746_{\pm 0.389}$ | $0.424_{\pm 0.049}$ | $0.206_{\pm 0.033}$ | $7.36_{\pm 2.43}$ | $0.23_{\pm 0.05}$ | $471.64_{\pm 1.20}$ | $489.30_{\pm 0.62}$ |
| PIS [88] | $1.769_{\pm 0.104}$ | $1.274_{\pm 0.218}$ | $0.534_{\pm 0.008}$ | $0.262_{\pm 0.008}$ | $3.85_{\pm 0.03}$ | $2.69_{\pm 0.04}$ | $381.14_{\pm 1.42}$ | $414.42_{\pm 2.06}$ |
| + LP [88] | $1.799_{\pm 0.051}$ | $0.225_{\pm 0.583}$ | $0.587_{\pm 0.012}$ | $0.285_{\pm 0.044}$ | $13.19_{\pm 0.82}$ | $0.07_{\pm 0.85}$ | $471.45_{\pm 0.18}$ | $487.82_{\pm 2.26}$ |
| TB [42] | $1.176_{\pm 0.109}$ | $1.071_{\pm 0.112}$ | $0.690_{\pm 0.018}$ | $0.239_{\pm 0.192}$ | $4.01_{\pm 0.04}$ | $2.67_{\pm 0.02}$ | $336.70_{\pm 56.22}$ | $379.50_{\pm 49.99}$ |
| TB + Expl. [42] | $0.560_{\pm 0.302}$ | $0.422_{\pm 0.320}$ | $0.749_{\pm 0.015}$ | $0.226_{\pm 0.138}$ | $4.01_{\pm 0.05}$ | $2.68_{\pm 0.06}$ | $346.10_{\pm 55.54}$ | $389.21_{\pm 44.13}$ |
| VarGrad + Expl. | $0.615_{\pm 0.241}$ | $0.487_{\pm 0.250}$ | $0.642_{\pm 0.010}$ | $0.250_{\pm 0.112}$ | $4.01_{\pm 0.05}$ | $2.69_{\pm 0.06}$ | $370.37_{\pm 0.26}$ | $410.37_{\pm 6.70}$ |
| FL-SubTB | $1.127_{\pm 0.010}$ | $1.020_{\pm 0.010}$ | $0.527_{\pm 0.011}$ | $0.182_{\pm 0.142}$ | $3.98_{\pm 0.07}$ | $2.72_{\pm 0.05}$ | $365.20_{\pm 6.08}$ | $402.65_{\pm 8.36}$ |
| + LP [86] | $0.209_{\pm 0.025}$ | $0.011_{\pm 0.024}$ | $0.563_{\pm 0.021}$ | $0.155_{\pm 0.317}$ | $4.23_{\pm 0.12}$ | $2.66_{\pm 0.22}$ | $465.44_{\pm 1.26}$ | $483.90_{\pm 1.95}$ |
| TB + Expl. + LS (*ours*) | $0.171_{\pm 0.013}$ | $0.004_{\pm 0.011}$ | $0.653_{\pm 0.025}$ | $0.285_{\pm 0.099}$ | $4.57_{\pm 2.13}$ | $0.19_{\pm 0.29}$ | $384.90_{\pm 0.83}$ | $419.55_{\pm 2.14}$ |
| TB + Expl. + LP (*ours*) | $0.206_{\pm 0.018}$ | $0.011_{\pm 0.010}$ | $0.666_{\pm 0.615}$ | $0.051_{\pm 0.616}$ | $7.46_{\pm 1.74}$ | $1.06_{\pm 1.11}$ | $452.82_{\pm 1.50}$ | $477.62_{\pm 1.79}$ |
| TB + Expl. + LP + LS (*ours*) | $0.190_{\pm 0.013}$ | $0.007_{\pm 0.011}$ | $0.768_{\pm 0.052}$ | $0.264_{\pm 0.063}$ | $4.68_{\pm 0.49}$ | $0.07_{\pm 0.17}$ | $471.14_{\pm 0.25}$ | $489.03_{\pm 1.38}$ |
| VarGrad + Expl. + LP + LS (*ours*) | $0.207_{\pm 0.016}$ | $0.015_{\pm 0.015}$ | $0.920_{\pm 0.118}$ | $0.256_{\pm 0.037}$ | $4.11_{\pm 0.45}$ | $0.02_{\pm 0.21}$ | $468.65_{\pm 0.63}$ | $487.34_{\pm 1.34}$ |

Highlight : mean indistinguishable from best in column with $p < 0.05$ under one-sided Welch unpaired $t$-test.

To enhance the quality of samples during training, we incorporate local search into the exploration process, motivated by the success of local exploration [83, 33, 40] and replay buffer [*e.g.*, 17] methods for GFlowNets in discrete spaces. Unlike these methods, which define MCMC kernels via the GFlowNet policies, our method leverages parallel Metropolis-adjusted Langevin (MALA) directly in the target space.

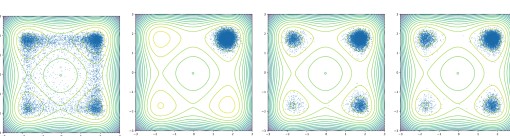

PIS + LP     TB + Expl.     TB + Expl. + LS     True samples

Figure 1: Two-dimensional projections of **Manywell** samples from models trained by different algorithms. Our proposed replay buffer with local search is capable of preventing mode collapse.

In detail, we initially sample $M$ candidates from the sampler: $\{\mathbf{x}^{(1)}, \ldots, \mathbf{x}^{(M)}\} \sim p_F^\top(\cdot)$. Subsequently, we run parallel MALA across $M$ chains over $K$ transitions , with the initial states of the Markov chain being $\{\mathbf{x}^{(1)}, \ldots, \mathbf{x}^{(M)}\}$. After the $K_{\mathrm{burn\text{-}in}}$ burn-in transitions, the accepted samples are stored in a local search buffer $\mathcal{D}_{\mathrm{LS}}$. We occasionally update the buffer using MALA steps and replay samples from it to minimize the computational demands of iterative local search. MALA steps are far more parallelizable than sampler training and need to be made only rarely (as the buffer is much larger than the training batch size), so the overhead of local search is small.

**Training with local search and replay buffer.** To train samplers with the aid of the buffer, we draw a sample $\mathbf{x}$ from $\mathcal{D}_{\mathrm{LS}}$ (uniformly or using a prioritization scheme, §E), sample a trajectory $\tau$ leading to $\mathbf{x}$ from the backward process, and make a gradient update on the objective (*e.g.*, TB) associated with $\tau$.

When training with local search guidance, we alternate two steps, inspired by [43], who alternate training on forward trajectories and backward trajectories initialized at a *fixed* set of MCMC samples. Step A involves training with on-policy or exploratory forward sampling while Step B uses samples drawn from the local search buffer described above. This allows the sampler to explore both diversified samples (Step A) and low-energy samples (Step B). See §E for detailed pseudocode of adaptive-step parallel MALA and local search-guided GFlowNet training.

## 5    Experiments

We conduct comprehensive benchmarks of various diffusion-structured samplers, encompassing both GFlowNet samplers and methods such as PIS. For the GFlowNet samplers, we investigate a range of techniques, including different exploration strategies and loss functions. Additionally, we examine the efficacy of the Langevin parametrization and the newly proposed local search with buffer.

### 5.1    Tasks and baselines

We explore two types of tasks, with more details provided in §B: sampling from energy distributions – a 2-dimensional mixture of Gaussians with 25 modes (**25GMM**), the 10-dimensional **Funnel**, the 32-dimensional **Manywell** distribution, and the 1600-dimensional **Log-Gaussian Cox process** –

and *conditional* sampling from the latent posterior of a variational autoencoder (**VAE**; [41, 61]). This allows us to investigate both unconditional and conditional generative modeling techniques.

We evaluate three algorithm categories:

(1) **Traditional sampling methods:** We consider a standard Sequential Monte Carlo (SMC) implementation and a state-of-the-art nested sampling method (GGNS, [43]).
(2) **Simulation-driven variational approaches:** DIS [8], DDS [78], and PIS [88].
(3) **Diffusion-based GFlowNet samplers:** Our evaluation focuses on TB-based training and the enhancements described in §4: the VarGrad estimator (**VarGrad**), off-policy exploration (**Expl.**), Langevin parametrization (**LP**), and local search (**LS**). Additionally, we assess the FL-SubTB-based continuous GFlowNet as studied by [86] for a comprehensive comparison.

For (2) and (3), we employ a consistent neural architecture across methods (details in §D).

**Learning problem and fixed backward process.** In our main experiments, we borrow the modeling setting from [88]. We aim to learn a Gaussian forward policy $p_F$ that samples from the target distribution in $T = 100$ steps ($\Delta t = 0.01$). Just as in past work [88, 42, 86], the backward process is fixed to a discretized Brownian bridge with a noise rate $\sigma$ that depends on the domain; explicitly,

$$p_B(\mathbf{x}_{t-\Delta t} \mid \mathbf{x}_t) = \mathcal{N}\left(\mathbf{x}_{t-\Delta t}; \frac{t - \Delta t}{t}\mathbf{x}_t, \frac{t - \Delta t}{t}\sigma^2 \Delta t \mathbf{I}_d\right), \tag{15}$$

understood to be a point mass at $\mathbf{0}$ when $t = \Delta t$. To keep the learning problem consistent with past work, we fix the variance of the forward policy $p_F$ to $\sigma^2$. This simplification is justified in continuous time, when the forward and reverse SDEs have the same diffusion rate. However, in §5.3, we will provide evidence that *learning* the forward policy's variance is quite beneficial for shorter trajectories.

**Benchmarking metrics.** To evaluate diffusion-based samplers, we use two metrics from past work [88, 42], which we restate in our notation. Given any forward policy $p_F$, we have a variational lower bound on the log-partition function $\log Z = \int_{\mathbb{R}^d} R(\mathbf{x})\, d\mathbf{x}$:

$$\log \int_{\mathbb{R}^d} R(\mathbf{x})\, d\mathbf{x} = \log \mathop{\mathbb{E}}_{\tau = (\cdots \to \mathbf{x}_1) \sim p_F(\tau)} \left[ \frac{R(\mathbf{x}_1) p_B(\tau \mid \mathbf{x}_1)}{p_F(\tau)} \right] \geq \mathop{\mathbb{E}}_{\tau = (\cdots \to \mathbf{x}_1) \sim p_F(\tau)} \left[ \log \frac{R(\mathbf{x}_1) p_B(\tau \mid \mathbf{x}_1)}{p_F(\tau)} \right].$$

We use a $K$-sample ($K = 2000$) Monte Carlo estimate of this expectation, $\log \hat{Z}$, as a metric, which equals the true $\log Z$ if $p_F$ and $p_B$ jointly satisfy (10) and thus $p_F$ samples from the target distribution. We also employ an importance-weighted variant, which emphasizes mode coverage over accurate local modeling:

$$\log \hat{Z}^{\mathrm{RW}} := \log \sum_{i=1}^{K} \left[ \frac{R(\mathbf{x}_1^{(i)}) p_B(\tau^{(i)} \mid \mathbf{x}_1^{(i)})}{p_F(\tau^{(i)})} \right],$$

where $\tau^{(1)}, \ldots, \tau^{(K)}$ are trajectories sampled from $p_F$ and leading to terminal states $\mathbf{x}_1^{(1)}, \ldots, \mathbf{x}_1^{(K)}$. The estimator $\log \hat{Z}^{\mathrm{RW}}$ is also a lower bound on $\log Z$ and approaches it as $K \to \infty$ [11]. In the unconditional modeling benchmarks, we compare both estimators to the true log-partition function, which is known analytically for all tasks except LGCP (leading to discrepancies in past work; see §B.1).

In addition, we include a sample-based metric (2-Wasserstein distance); see §C.1.

## 5.2 Results

**Unconditional sampling.** We report the metrics for all algorithms and energies in Table 1.

We observe that TB's performance is generally modest without additional exploration and credit assignment mechanisms, except on the **Funnel** task, where variations in performance across methods are negligible. This confirms hypotheses from past work about the importance of off-policy exploration [46, 42] and the importance of improved credit assignment [86]. On the other hand, our results do not show a consistent

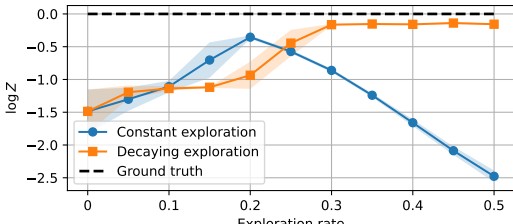

Figure 2: Effect of exploration variance on models trained with TB on the **25GMM** energy. Exploration promotes mode discovery, but should be decayed over time to optimally allocate the modeling power to high-likelihood trajectories.

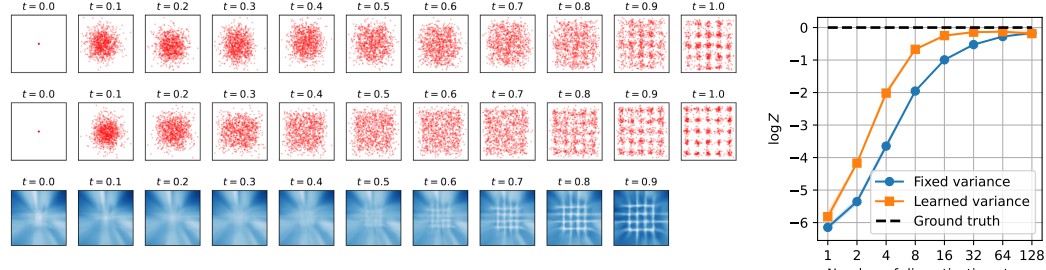

Figure 3: **Left:** Distribution of $\mathbf{x}_0, \mathbf{x}_{0.1}, \ldots, \mathbf{x}_1$ learned by 10-step samplers with fixed (*top*) and learned (*middle*) forward policy variance on the **25GMM** energy. The last step of sampling the *fixed-variance* model adds Gaussian noise of a variance close to that of the components of the target distribution, preventing the the sampler from sharply capturing the modes. The last row shows the policy variance learned as a function of $\mathbf{x}_t$ at various time steps $t$ (white is high variance, blue is low), showing that less noise is added around the peaks near $t = 1$. The two models' log-partition function estimates are $-1.67$ and $-0.62$, respectively. **Right:** For varying number of steps $T$, we plot the $\log \hat{Z}$ obtained by models with fixed and learned variance. **Learning policy variances gives similar samplers with fewer steps.**

and significant improvement of the FL-SubTB
objective used by [86] over TB. Replacing TB with the VarGrad objective yields similar results.

The simple off-policy exploration method of adding variance to the policy notably enhances performance on the **25GMM** task. We investigate this phenomenon in more detail in Fig. 2, finding that exploration that slowly decreases over the course of training is the best strategy.

On the other hand, our local search-guided exploration with a replay buffer (LS) leads to a substantial improvement in performance, surpassing or competing with GFlowNet baselines, non-GFlowNet baselines, and non-amortized sampling methods in most tasks and metrics. This advantage is attributed to efficient exploration and the ability to replay past low-energy regions, thus preventing mode collapse during training (Fig. 1). Further details on LS enhancements are discussed in §E with ablation studies in §E.2.

Incorporating Langevin parametrization (LP) into TB or FL-SubTB results in notable performance improvements (despite being 2-3× slower per iteration), indicating that previous observations [88] transfer to off-policy algorithms. Compared to FL-SubTB, which aims for enhanced credit assignment through partial energy, LP achieves superior credit assignment leveraging gradient information, akin to partial energy in continuous time. LP is either superior or competitive across most tasks and metrics.

In §C.3, we study the scaling of the algorithms with dimension, showing efficiency of the proposed LS.

**Conditional sampling.** For the VAE task, we observe that the performance of the baseline GFlowNet-based samplers is generally worse than that of the simulation-based PIS (Table 2). While LP and LS improve the performance of

Table 2: Log-likelihood estimates on a test set for a pretrained VAE decoder on MNIST. The latent being sampled is 20-dimensional. The VAE's training ELBO (Gaussian encoder) was $\approx -101$.

| Algorithm ↓ Metric → | $\log \hat{Z}$ | $\log \hat{Z}^{\text{RW}}$ |
|---|---|---|
| GGNS [43] | $-82.406_{\pm 0.882}$ | |
| PIS [88] | $-102.54_{\pm 0.437}$ | $-47.753_{\pm 2.821}$ |
| + LP [88] | $-99.890_{\pm 0.373}$ | $-47.326_{\pm 0.777}$ |
| TB [42] | $-162.73_{\pm 35.55}$ | $-61.407_{\pm 17.83}$ |
| VarGrad | $-102.54_{\pm 0.934}$ | $-46.502_{\pm 1.018}$ |
| TB + Expl. [42] | $-148.04_{\pm 4.046}$ | $-49.967_{\pm 5.683}$ |
| FL-SubTB | $-147.992_{\pm 22.671}$ | $-54.196_{\pm 3.996}$ |
| + LP [86] | $-111.536_{\pm 1.027}$ | $-47.640_{\pm 1.313}$ |
| TB + Expl. + LS (*ours*) | $-245.78_{\pm 13.80}$ | $-55.378_{\pm 9.125}$ |
| TB + Expl. + LP (*ours*) | $-112.45_{\pm 0.671}$ | $-48.827_{\pm 1.787}$ |
| TB + Expl. + LP + LS (*ours*) | $-117.26_{\pm 2.502}$ | $-49.157_{\pm 2.051}$ |
| VarGrad + Expl. (*ours*) | $-103.39_{\pm 0.691}$ | $-47.318_{\pm 1.981}$ |
| VarGrad + Expl. + LS (*ours*) | $-105.40_{\pm 0.882}$ | $-48.235_{\pm 0.891}$ |
| VarGrad + Expl. + LP (*ours*) | $-99.472_{\pm 0.259}$ | $-46.574_{\pm 0.736}$ |
| VarGrad + Expl. + LP + LS (*ours*) | $-99.783_{\pm 0.312}$ | $-46.245_{\pm 0.543}$ |

TB, they do not close the gap in likelihood estimation; however, with the VarGrad objective, the performance is competitive with or superior to PIS. We hypothesize that this discrepancy is due to the difficulty of fitting the conditional log-partition function estimator, which is required for the TB objective but not for VarGrad, which only learns the policy. (In Fig. D.1 we show decoded samples encoded using the best-performing diffusion encoder.)

## 5.3 Extensions to general SDE learning problems

Our implementation of diffusion-structured generative flow networks includes several additional options that diverge from the modeling assumptions made in most past work in the field. Notably, it features the ability to:

- **optimize the backward (noising) process** – not only the denoising process – as was done for related learning problems in [12, 63, 79];
- **learn the forward process's diffusion rate** $g(\mathbf{x}_t, t; \theta)$, not only the mean $u(\mathbf{x}_t, t; \theta)$;
- assume a **varying noise schedule** for the backward process, making it possible to train models with standard noising SDEs used for diffusion models for images.

These extensions will allow others to build on our implementation and apply it to problems such as finetuning diffusion models trained on images with a GFlowNet objective.

As noted in §5.1, in the main experiments we fixed the diffusion rate of the learned forward process, an assumption inherited from all past work and justified in the continuous-time limit. However, we perform an experiment to show the importance of extensions such as learning the forward variance in discrete time. Fig. 3 shows the samples of models on the **25GMM** energy following the experimental setup of [43]. We see that when the forward policy's variance is learned, the model can better capture the details of the target distributions, choosing a low variance in the vicinity of the peaks to avoid 'blurring' them through the noise added in the last step of sampling.

In §C.2, we include preliminary results using a variance-preserving backward process, as commonly used in diffusion models, in place of the reversed Brownian motion used in the main experiments.

The ability to model distributions accurately in fewer steps is important for computational efficiency. Future work can consider ways to improve performance in coarse time discretizations, such as non-Gaussian transitions, whose utility in diffusion models trained from data has been demonstrated [82].

## 6 Conclusion

We have presented a study of diffusion-structured samplers for amortized inference over continuous variables. Our results suggest promising techniques for improving the mode coverage and efficiency of these models. Future work on applications can consider inference of high-dimensional parameters of dynamical systems and inverse problems. In probabilistic machine learning, extensions of this work should study integration of our amortized sequential samplers as variational posteriors in an expectation-maximization loop for training latent variable models, as was recently done for discrete compositional latents by [33], and for sampling Bayesian posteriors over high-dimensional model parameters. The most important direction of theoretical work is understanding the continuous-time limit ($T \rightarrow \infty$) of all the algorithms we have studied.

*Note added in final version:* In a paper that appeared subsequently to the publication of this work, Berner et al. [9] have shown connections among the families of diffusion sampling algorithms considered here and analyzed their continuous-time limits.

## Acknowledgments

We thank Cheng-Hao Liu for assistance with methods from prior work, as well as Julius Berner, Víctor Elvira, Lorenz Richter, Alexander Tong, and Siddarth Venkatraman for helpful discussions and suggestions.

The authors acknowledge funding from UNIQUE, CIFAR, NSERC, Intel, Recursion Pharmaceuticals, and Samsung. The research was enabled in part by computational resources provided by the Digital Research Alliance of Canada (https://alliancecan.ca), Mila (https://mila.quebec), and NVIDIA. The research of M.S. was in part funded by National Science Centre, Poland, 2022/45/N/ST6/03374.

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

# A  Code and hyperparameters

Code is available at  and will continue to be maintained and extended.

Below are commands to reproduce some of the results on **Manywell** and **VAE** with PIS and GFlowNet models as an example, showing the hyperparameters:

PIS:

```
--mode_fwd pis  --lr_policy 1e-3
```

PIS + Langevin:

```
--mode_fwd pis  --lr_policy 1e-3 --langevin
```

GFlowNet TB:

```
python train.py
--t_scale 1. --energy many_well --pis_architectures --zero_init --clipping
--mode_fwd tb --lr_policy 1e-3 --lr_flow 1e-1
```

GFlowNet TB + Expl.:

```
python train.py
--t_scale 1. --energy many_well --pis_architectures --zero_init --clipping
--mode_fwd tb --lr_policy 1e-3 --lr_flow 1e-1
--exploratory --exploration_wd --exploration_factor 0.2
```

GFlowNet VarGrad + Expl.:

```
python train.py
--t_scale 1. --energy many_well --pis_architectures --zero_init --clipping
--mode_fwd tb-avg --lr_policy 1e-3 --lr_flow 1e-1
--exploratory --exploration_wd --exploration_factor 0.2
```

GFlowNet FL-SubTB:

```
python train.py
--t_scale 1. --energy many_well --pis_architectures --zero_init --clipping
--mode_fwd subtb --lr_policy 1e-3 --lr_flow 1e-2
--partial_energy --conditional_flow_model
```

GFlowNet FL-SubTB + LP:

```
python train.py
--t_scale 1. --energy many_well --pis_architectures --zero_init --clipping
--mode_fwd subtb --lr_policy 1e-3 --lr_flow 1e-2
--partial_energy --conditional_flow_model
--langevin --epochs 10000
```

GFlowNet TB + Expl. + LS:

```
python train.py
--t_scale 1. --energy many_well --pis_architectures --zero_init --clipping
--mode_fwd tb --lr_policy 1e-3 --lr_back 1e-3 --lr_flow 1e-1
--exploratory --exploration_wd --exploration_factor 0.1
--both_ways --local_search
--buffer_size 600000 --prioritized rank --rank_weight 0.01
--ld_step 0.1 --ld_schedule --target_acceptance_rate 0.574
```

GFlowNet TB + Expl. + LP:

```
python train.py
--t_scale 1. --energy many_well --pis_architectures --zero_init --clipping
--mode_fwd tb --lr_policy 1e-3 --lr_flow 1e-1
--exploratory --exploration_wd --exploration_factor 0.2
--langevin --epochs 10000
```

GFlowNet TB + Expl. + LS (VAE):

```
python train.py
--energy vae --pis_architectures --zero_init --clipping
--mode_fwd cond-tb-avg --mode_bwd cond-tb-avg --repeats 5
--lr_policy 1e-3 --lr_flow 1e-1 --lr_back 1e-3
--exploratory --exploration_wd --exploration_factor 0.1
--both_ways --local_search
--max_iter_ls 500 --burn_in 200
--buffer_size 90000 --prioritized rank --rank_weight 0.01
--ld_step 0.001 --ld_schedule --target_acceptance_rate 0.574
```

GFlowNet TB + Expl. + LP + LS (VAE):

```
python train.py
--energy vae --pis_architectures --zero_init --clipping
--mode_fwd cond-tb-avg --mode_bwd cond-tb-avg --repeats 5
--lr_policy 1e-3 --lr_flow 1e-1
--lgv_clip 1e2 --gfn_clip 1e4 --epochs 10000
--exploratory --exploration_wd --exploration_factor 0.1
--both_ways --local_search
--lr_back 1e-3 --max_iter_ls 500 --burn_in 200
--buffer_size 90000 --prioritized rank  --rank_weight 0.01
--langevin
--ld_step 0.001 --ld_schedule --target_acceptance_rate 0.574
```

# B    Target densities

**Gaussian Mixture Model with 25 modes (25GMM).** The model, termed as 25GMM, consists of a two-dimensional Gaussian mixture model with 25 distinct modes. Each mode exhibits an identical variance of 0.3. The centers of these modes are strategically positioned on a grid formed by the Cartesian product $\{-10, -5, 0, 5, 10\} \times \{-10, -5, 0, 5, 10\}$, effectively distributing them across the coordinate space.

**Funnel [29].** The funnel represents a classical benchmark in sampling techniques, characterized by a ten-dimensional distribution defined as follows: The first dimension, $x_0$, follows a normal distribution with mean 0 and variance 9, denoted as $x_0 \sim \mathcal{N}(0, 9)$. Conditional on $x_0$, the remaining dimensions, $x_{1:9}$, are distributed according to a multivariate normal distribution with mean vector $\mathbf{0}$ and a covariance matrix $\exp(x_0)\mathbf{I}$, where $\mathbf{I}$ is the identity matrix. This is succinctly represented as $x_{1:9} \mid x_0 \sim \mathcal{N}(\mathbf{0}, \exp(x_0)\mathbf{I})$.

**Manywell [52].** The manywell is characterized by a 32-dimensional distribution, which is constructed as the product of 16 identical two-dimensional double well distributions. Each of these two-dimensional components is defined by a potential function, $\mu(x_1, x_2)$, expressed as $\mu(x_1, x_2) = \exp\left(-x_1^4 + 6x_1^2 + 0.5x_1 - 0.5x_2^2\right)$.

**VAE [41].** This task involves sampling from a 20-dimensional latent posterior $p(z|x) \propto p(z)p(x|z)$, where $p(z)$ is a fixed prior and $p(x|z)$ is a pretrained VAE decoder, using a conditional sampler $q(z|x)$ dependent on input data (image) $x$.

**LGCP [49].** This density over a 1600-dimensional variable is a Log-Gaussian Cox process fit to a distribution of pine saplings in Finland.

## B.1    Discrepancies in past work

**Wrong definitions of the Funnel density.** As already noted by [78], [88] uses a different variance of the first component in the Funnel density, 1 instead of 9. This apparent bug in the task definition has been propagated to subsequent work, including [42].

**Evaluation on LGCP.** The LGCP benchmark suffers from the lack of a consistent ground truth $\log Z$ to compare against. Previous work has compared the value of the partition function $\log Z$ against a "long run of Sequential Monte Carlo" [88]. We note that this approach produces noisy estimates of the partition function, especially in high-dimensional problems (indeed, SMC has rarely been used in problems with over a thousand dimensions); therefore, it is unclear how long the SMC needs to be run to produce an accurate estimate. We found that two *different* values are being used in the literature: $\log Z = 512.6$ in one repository and $\log Z = 501.8$ in another.

**On FL-SubTB as used in [86].** We make two observations calling into question the main results of [86].

First, the only substantial difference between the algorithm used by [86] and the one from the past work [42] – which first proposed the use of GFlowNet objectives to train diffusion samplers – is the substitution of the FL-SubTB objective [55, 44] for TB [45]. However, [86] elects to compare FL-SubTB *with* the Langevin parameterization to TB *without* the Langevin parameterization. Our results in Table 1 show that while the Langevin parameterization is crucial for the performance of all objectives; FL-SubTB does not provide any consistent benefit over TB or VarGrad.

Second, the results are not reproducible, neither with the published code from [86] run 'out of the box', nor with our reimplementation. In particular, on the LGCP density, the training did not converge within the allotted training time. We have contacted the authors of [86], who confirmed that running their published code does not reproduce the results in the paper but could not provide any further explanation or a working implementation.

# C    Additional results

## C.1    Expanded unconditional sampling results

Table C.1 is an expanded version of Table 1, showing Wasserstein distances between sets of $K$ samples from the true distribution and generated by a trained sampler. (Note that ground truth for LGCP is not available.)

Table C.1: Log-partition function estimation errors and 2-Wasserstein distances for unconditional modeling tasks (mean and standard deviation over 5 runs). The four groups of models are: MCMC-based samplers, simulation-driven variational methods, baseline GFlowNet methods with different learning objectives, and methods augmented with Langevin parametrization and local search.

| Energy → | 25GMM ($d = 2$) | | | Funnel ($d = 10$) | | | Manywell ($d = 32$) | | |
|---|---|---|---|---|---|---|---|---|---|
| Algorithm ↓ Metric → | $\Delta \log Z$ | $\Delta \log Z^{\mathrm{RW}}$ | $\mathcal{W}_2^2$ | $\Delta \log Z$ | $\Delta \log Z^{\mathrm{RW}}$ | $\mathcal{W}_2^2$ | $\Delta \log Z$ | $\Delta \log Z^{\mathrm{RW}}$ | $\mathcal{W}_2^2$ |
| SMC | 0.569±0.010 | | 0.86±0.10 | 0.561±0.801 | | 50.3±18.9 | 14.99±1.078 | | 8.28±0.32 |
| GGNS [43] | 0.016±0.042 | | 1.19±0.17 | 0.033±0.173 | | 25.6±4.75 | 0.292±0.454 | | 6.51±0.32 |
| DIS [8] | 1.125±0.056 | 0.986±0.011 | 4.71±0.06 | 0.839±0.169 | 0.093±0.038 | 20.7±2.1 | 10.52±1.02 | 3.05±0.46 | 5.98±0.46 |
| DDS [78] | 1.760±0.08 | 0.746±0.389 | 7.18±0.044 | 0.424±0.049 | 0.206±0.033 | 29.3±9.5 | 7.36±2.43 | 0.23±0.05 | 5.71±0.16 |
| PIS [88] | 1.769±0.104 | 1.274±0.218 | 6.37±0.65 | 0.534±0.008 | 0.262±0.008 | 22.0±4.0 | 3.85±0.03 | 2.69±0.04 | 6.15±0.02 |
| + LP [88] | 1.799±0.051 | 0.225±0.583 | 7.16±0.11 | 0.587±0.012 | 0.285±0.044 | 22.1±4.0 | 13.19±0.82 | 0.07±0.85 | 6.55±0.34 |
| TB [42] | 1.176±0.109 | 1.071±0.112 | 4.83±0.45 | 0.690±0.018 | 0.239±0.192 | 22.4±4.0 | 4.01±0.04 | 2.67±0.02 | 6.14±0.02 |
| TB + Expl. [42] | 0.560±0.302 | 0.422±0.320 | 3.61±1.41 | 0.749±0.015 | 0.226±0.138 | 21.3±4.0 | 4.01±0.05 | 2.68±0.06 | 6.15±0.02 |
| VarGrad + Expl. | 0.615±0.241 | 0.487±0.250 | 3.89±0.85 | 0.642±0.010 | 0.250±0.112 | 22.1±4.0 | 4.01±0.05 | 2.69±0.06 | 6.15±0.02 |
| FL-SubTB | 1.127±0.010 | 1.020±0.010 | 4.64±0.09 | 0.527±0.011 | 0.182±0.142 | 22.1±4.0 | 3.98±0.07 | 2.72±0.05 | 6.15±0.01 |
| + LP [86] | 0.209±0.025 | 0.011±0.024 | 1.45±0.29 | 0.563±0.021 | 0.155±0.317 | 22.2±4.0 | 4.23±0.12 | 2.66±0.22 | 6.10±0.02 |
| TB + Expl. + LS (*ours*) | 0.171±0.013 | 0.004±0.011 | 1.25±0.18 | 0.653±0.025 | 0.285±0.099 | 21.9±4.0 | 4.57±2.13 | 0.19±0.29 | 5.66±0.05 |
| TB + Expl. + LP (*ours*) | 0.206±0.018 | 0.011±0.010 | 1.29±0.07 | 0.666±0.615 | 0.051±0.616 | 22.3±3.9 | 7.46±1.74 | 1.06±1.11 | 5.73±0.31 |
| TB + Expl. + LP + LS (*ours*) | 0.190±0.013 | 0.007±0.011 | 1.31±0.07 | 0.768±0.052 | 0.264±0.063 | 21.8±3.9 | 4.68±0.49 | 0.07±0.17 | 5.33±0.03 |
| VarGrad + Expl. + LP + LS (*ours*) | 0.207±0.016 | 0.015±0.015 | 1.13±0.13 | 0.920±0.118 | 0.256±0.037 | 21.2±4.0 | 4.11±0.45 | 0.02±0.21 | 5.30±0.02 |

Highlight : mean indistinguishable from minimum in column with $p < 0.05$ under one-sided Welch unpaired $t$-test.

Table C.2: Log-partition function estimation errors and empirical 2-Wasserstein distances on the 32-dimensional Manywell with Brownian and variance-preserving noising processes.

| Backward process → | Brownian | | | VP | | |
|---|---|---|---|---|---|---|
| Objective ↓ Metric → | $\Delta \log Z$ | $\Delta \log Z^{\mathrm{RW}}$ | $\mathcal{W}_2^2$ | $\Delta \log Z$ | $\Delta \log Z^{\mathrm{RW}}$ | $\mathcal{W}_2^2$ |
| TB + Expl. + LP | 7.46±1.74 | 1.06±1.11 | 5.73±0.31 | 7.55±2.85 | 1.49±1.30 | 5.68±0.42 |
| TB + Expl. + LP + LS | 4.68±0.49 | 0.07±0.17 | 5.33±0.03 | 4.52±0.21 | 1.23±0.07 | 5.75±0.01 |
| VarGrad + Expl. | 4.01±0.05 | 2.69±0.06 | 6.15±0.02 | 4.04±0.05 | 2.65±0.08 | 6.17±0.02 |

## C.2 Variance-preserving noising process

Following the recent results by [8, 63, 78], we perform an additional set of experiments with a different successful noise schedule. We replace the Brownian motion by the *variance-preserving SDEs* from Song et al. [72], given by an *Ornstein-Uhlenbeck process:*

$$\sigma(t) := \nu\sqrt{2\beta(t)}\mathbf{I} \quad \text{and} \quad \mu(x,t) := -\beta(t)x \tag{16}$$

with $\nu \in (0, \infty)$.

In particular, we follow the common procedure - use $\nu := 1$ and

$$\beta(t) := (1-t)\beta_{min} + t\beta_{max}, \quad t \in [0,1],$$

with $\beta_{min} = 0.01$ and $\beta_{max} = 4.0$.

We evaluate three representative methods using this variance-preserving backward process. The results, in Table C.2, are similar to those using the Brownian bridge process. We expect that the choice of noising process gains importance in challenging high-dimensional problems.

## C.3 Scalability study

The Manywell energy (§B) is defined in any even number of dimensions and thus allows to study the scaling of the methods with dimension. We evaluate several representative methods in dimension 8, 128, and 512 (in addition to the 32 studied in the main text). All experimental settings are kept the same as as for $d = 32$. Due to the large runtime, some runs in dimensions 128 and 512 had to be limited at 12 hours, while in dimensions 8 and 32 all run in under 3 hours on a RTX8000 GPU.

These results are shown in Table C.3. We observe:

- The overhead of the Langevin parametrization grows with dimension, but is critical to performance.
- The even higher overhead of FL-SubTB as used by [86].
- The relatively high efficiency and low overhead of our newly proposed local search.

Table C.3: Scaling with dimension on Manywell: log-partition function estimation errors and time per training iteration on a RTX8000 GPU.

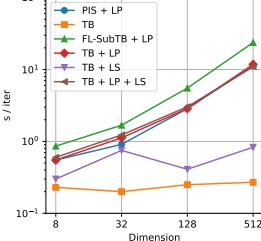

| Dimension → | $d=8$ | | $d=32$ | | $d=128$ | | $d=512$ | |
|---|---|---|---|---|---|---|---|---|
| Objective ↓ Metric → | $\Delta \log Z$ | $\Delta \log Z^{\mathrm{RW}}$ | $\Delta \log Z$ | $\Delta \log Z^{\mathrm{RW}}$ | $\Delta \log Z$ | $\Delta \log Z^{\mathrm{RW}}$ | $\Delta \log Z$ | $\Delta \log Z^{\mathrm{RW}}$ |
| PIS + LP [88] | 0.86 | 0.14 | 13.19 | 0.07 | 58.0 | 23.7 | 251 | 169 |
| TB [42] | 0.95 | 0.70 | 4.01 | 2.68 | 205.6 | 119.8 | 1223 | 957 |
| FL-SubTB + LP [86] | 0.57 | 0.67 | 4.23 | 2.66 | 48.9 | 21.7 | 198 | 107 |
| TB + LP | 0.25 | 0.04 | 7.46 | 1.06 | 46.4 | 14.0 | 259 | 169 |
| TB + LS | 0.44 | 0.00 | 4.57 | 0.19 | 458.7 | 139.3 | 1626 | 1077 |
| TB + LP + LS | 0.25 | 0.02 | 4.68 | 0.07 | 66.6 | 14.9 | 326 | 209 |

# D    Experiment details

**Sampling energies.** In this section, we detail the hyperparameters used for our experiments. An important parameter is the diffusion coefficient of the forward policy, which is denoted by $\sigma$ and also used in the definition of the fixed backward process. The base diffusion rate $\sigma^2$ (parameter `t_scale`) is set to 5 for **25GMM** and 1 for **Funnel** and **Manywell**, consistent with past work.

For **LGCP**, we found that using too small diffusion rate $\sigma^2$ (*e.g.,* $\sigma^2 = 1$) prevents the methods from achieving reasonable results. We tested different values of $\sigma^2 = \{1, 3, 5\}$, and selected $\sigma^2 = 5$, which gives the best results, which follows the findings in Zhang & Chen [88].

For all our experiments, we used a learning rate of $10^{-3}$. Additionally, we used a higher learning rate for learning the flow parameterization, which is set as $10^{-1}$ when using the TB loss and $10^{-2}$ with the SubTB loss. These settings were found to be consistently stable (unlike those with higher learning rates) and converge within the allotted number of steps (unlike those with lower learning rates).

For the SubTB loss, we experimented with the settings of $10\times$ lower learning rates for both flow and policy models communicated by the authors of [86], but found the results to be inferior both using their published code (and other unstated hyperparameters communicated by the authors) and using our reimplementation.

For models with exploration, we use an exploration factor of 0.2 (that is, noise with a variance of 0.2 is added to the policy when sampling trajectories for training), which decays linearly over the first half of training, consistent with [42].

We train all our models for $25,000$ iterations except those using Langevin dynamics, which are trained for $10,000$ iterations. This results in approximately equal computation time owing to the overhead from computation of the score at each sampling step.

We use the same neural network architecture for the GFlowNet as one of our baselines [88]. Similar to [88], we also use an initialization scheme with last-layer weights set to 0 at the start of training. Since the SubTB requires the flow function to be conditioned on the current state $\mathbf{x}_t$ and time $t$, we follow [86] and parametrize the flow model with the same architecture as the Langevin scaling model $\mathrm{NN}_2$ in [88]. Additionally, we perform clipping on the output of the network as well as the score obtained from the energy function, typically setting the clipping parameter of Langevin scaling model to $10^2$ and policy network to $10^4$, similarly to [78]:

$$f_\theta(k, x) = \mathrm{clip}\Big(\mathrm{NN}_1(k, x; \theta) + \mathrm{NN}_2(k; \theta) \odot \mathrm{clip}\big(\nabla \ln \pi(x), -10^2, 10^2\big), -10^4, 10^4\Big). \quad (17)$$

All models were trained with a batch size of 300. In each experiment, we train models on a single NVIDIA A100-Large GPU, if not stated explicitly otherwise.

**VAE experiment.** In the VAE experiment, we used a standard VAE model pretrained for 100 epochs on the MNIST dataset. The encoder $q(z|x)$ contains an input linear layer (784 neurons) followed by hidden linear layer (400 neurons), ReLU activation function, and two linear heads (20 neurons each) whose outputs were reparametrized to be means and scales of multivariate Normal distribution. The decoder consists of 20-dimensional input, one hidden layer (400 neurons), followed by the ReLU activation, and 784-dimensional output. The output is processed by the sigmoid function to be scaled properly into $[0, 1]$.

The goal is to sample conditionally on $x$ the latent $z$ from the unnormalized density $p(z, x) = p(z)p(x \mid z)$ (where $p(z)$ is the prior and $p(x|z)$ is the likelihood computed from the decoder), which

is proportional to the posterior $p(z \mid x)$. We reuse the model architectures from the unconditional sampling experiments, but also provide $x$ as an input to the first layer of the models expressing the policy drift (as well as the flow, for FL-SubTB) and add one hidden layer to process high-dimensional conditions. For models trained with TB, $\log Z_\theta$ also becomes a MLP taking $x$ as input.

The VarGrad and LS techniques require adaptations in the conditional setting. For LS, buffers ($\mathcal{D}_{\text{buffer}}$ and $\mathcal{D}_{\text{LS}}$) must store the associated conditions $x$ together with the samples $z$ and the corresponding unnormalized density $R(z; x)$, *i.e.*, a tuple of $(x, z, R(z; x))$. For VarGrad, because the partition function depends on the conditioning information $x$, it is necessary to compute variance over many trajectories sharing the same condition. We choose to sample 10 trajectories for each condition occurring in a minibatch and compute the VarGrad loss for each such set of 10 trajectories.

The VAE model was trained on the entire MNIST training set and never updated on the test part of MNIST. In order to evaluate samplers (with respect to the variational lower bound) on a unique set of examples, we chose the first 100 elements of MNIST test data. All of the samplers were trained having access to the MNIST training data and the frozen VAE decoder. For a fair comparison, samplers utilizing the LP were trained for $10,000$, whereas the remaining for $25,000$ iterations. In each iteration, a batch of 300 examples from MNIST was given as conditions. In each experiment, we train models on a single NVIDIA A100-Large GPU, if not stated explicitly otherwise.

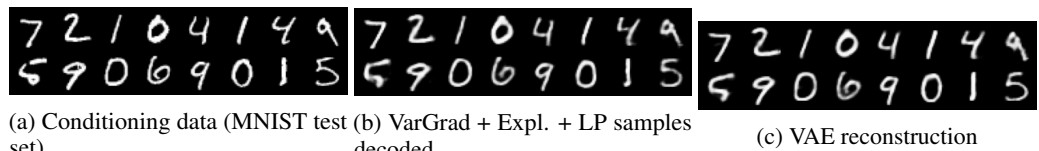

(a) Conditioning data (MNIST test set)
(b) VarGrad + Expl. + LP samples decoded
(c) VAE reconstruction

Figure D.1: Our sampler (VarGrad + Expl. + LP) is conditioned by a subset of never-seen data coming from the ground truth distribution (**left**). The conditional samples were then decoded by the the fixed VAE (**middle**). For the comparison, we show the reconstruction of the real data by VAE (**right**). We observed that the decoded samples are visually very similar to the reconstructions making these two pictures almost indistinguishable. Both, decoded samples and reconstruction, are more blurry than the ground truth data, which is caused by a limited capacity of the VAE's latent space.

# E   Local search-guided GFlowNet

**Prioritized sampling scheme.** We can use uniform or prioritized sampling to draw samples from the buffer for training. We found prioritized sampling to work slightly better in our experiments (see ablation study in §E.2), although the choice should be investigated more thoroughly in future work.

We use rank-based prioritization [74], which follows a probabilistic approach defined as:

$$p(\mathbf{x}; \mathcal{D}_{\text{buffer}}) \propto \left( k |\mathcal{D}_{\text{buffer}}| + \text{rank}_{\mathcal{D}_{\text{buffer}}}(\mathbf{x}) \right)^{-1}, \tag{18}$$

where $\text{rank}_{\mathcal{D}_{\text{buffer}}}(\mathbf{x})$ represents the relative rank of a sample $x$ based on a ranking function $R(\mathbf{x})$ (in our case, the unnormalized target density at sample $\mathbf{x}$). The parameter $k$ is a hyperparameter for prioritization, where a lower value of $k$ assigns a higher probability to samples with higher ranks, thereby introducing a more greedy selection approach. We set $k = 0.01$ for every task. Given that the sampling is proportional to the size of $\mathcal{D}_{\text{buffer}}$, we impose a constraint on the maximum size of the buffer: $|\mathcal{D}_{\text{buffer}}| = 600,000$ with first-in first out (FIFO) data structure for every task, except we use $|\mathcal{D}_{\text{buffer}}| = 90,000$ for VAE task. See the algorithm below for a detailed pseudocode.

---

**Algorithm 1** GFlowNet Training with Local search

---

1: Initialize policy parameters $\theta$ for $P_F$, and empty buffers $\mathcal{D}_{\text{buffer}}, \mathcal{D}_{\text{LS}}$
2: **for** $i = 1, 2, \ldots, I$ **do**
3:     **if** $i\%2 == 0$ **then**
4:         Sample $M$ trajectories $\{\tau_1, \ldots, \tau_M\} \sim P_F(\cdot | \epsilon\text{-greedy})$
5:         Update $\mathcal{D}_{\text{buffer}} \leftarrow \mathcal{D}_{\text{buffer}} \cup \{x | \tau \rightarrow x\}$
6:         Minimize $L(\tau; \theta)$ using $\{\tau_1, \ldots, \tau_M\}$ to update $P_F$
7:     **else**
8:         **if** $i\%100 == 0$ **then**
9:             Sample $\{x_1, \ldots, x_M\} \sim \mathcal{D}_{\text{buffer}}$
10:             $\mathcal{D}_{\text{LS}} \leftarrow \text{Local Search}(\{x_1, \ldots, x_M\}; \mathcal{D}_{\text{LS}})$
11:         **end if**
12:         Sample $\{x'_1, \ldots, x'_M\} \sim p_{\text{buffer}}(\cdots ; \mathcal{D}_{\text{LS}})$
13:         Sample $\{\tau'_1, \ldots, \tau'_M\} \sim P_B(\cdots | x')$
14:         Minimize $L(\tau'; \theta)$ using $\{\tau'_1, \ldots, \tau'_M\}$ to update $P_F$
15:     **end if**
16: **end for**

---

We use the number of total iterations $I = 25,000$ for every task as default. Note as local search is performed to update $\mathcal{D}_{\text{LS}}$ occasionally that per 100 iterations, the number of local search updates is done $25,000/100 = 250$.

### E.1 Local search algorithm

This section describes a detailed algorithm for local search, which provides an updated buffer $\mathcal{D}_{\text{LS}}$, which contains low-energy samples.

**Dynamic adjustment of step size $\eta$.** To enhance local search using parallel MALA, we dynamically select the Langevin step size ($\eta$), which governs the MH acceptance rate. Our objective is to attain an average acceptance rate of 0.574, which is theoretically optimal for high-dimensional MALA's efficiency [56]. While the user can customize the target acceptance rate, the adaptive approach eliminates the need for manual tuning.

**Computational cost of local search.** The computational cost of local search is not significant. Local search for iteration of $K = 200$ requires 6.04 seconds (averaged with five trials in Manywell), where we only occasionally (every 100 iterations) update $\mathcal{D}_{\text{LS}}$ with MALA. The speed is evaluated using the computational resources of the Intel Xeon Scalable Gold 6338 CPU (2.00GHz) and the NVIDIA RTX 4090 GPU.

---

**Algorithm 2** Local search (Parallel MALA)

---

**input** Initial states $\{x_1^{(0)}, \ldots, x_M^{(0)}\}$, current buffer $\mathcal{D}_{\text{LS}}$, total steps $K$, burn in steps $K_{\text{burn-in}}$, initial step size $\eta_0$, amplifying factor $f_{\text{increase}}$, damping factor $f_{\text{decrease}}$, unnormalized target density $R$
**output** Updated buffer $\mathcal{D}_{\text{LS}}$
  Initialize acceptance counter $a = 0$
  Set $\eta \leftarrow \eta_0$
  **for** $k = 1 : K$ **do**
    Initialize step acceptance count $a_k = 0$
    **for** $m = 1 : M$ **do**
      Sample $\sigma \sim \mathcal{N}(0, I)$
      Propose $x_m^* \leftarrow x_m^{(k-1)} + \eta \nabla \log R(x_m^{(k-1)}) + \sqrt{2\eta}\sigma$
      Compute acceptance ratio $r \leftarrow \min\left(1, \dfrac{R(x_m^*)\exp\left(-\frac{1}{4\eta}\|x_m^{(k-1)} - x_m^* - \eta \nabla \log R(x_m^*)\|^2\right)}{R(x_m^{(k-1)})\exp\left(-\frac{1}{4\eta}\|x_m^* - x_m^{(k-1)} - \eta \nabla \log R(x_m^{(k-1)})\|^2\right)}\right)$
      With probability $r$, accept the proposal: $x_m^{(k)} \leftarrow x_m^*$ and increment $a_k \leftarrow a_k + 1$
      **if** $k > K_{\text{burn-in}}$ **then**
        Update buffer: $\mathcal{D}_{\text{LS}} \leftarrow \mathcal{D}_{\text{LS}} \cup \{x_m^*\}$
      **end if**
    **end for**
    Compute step acceptance rate $\alpha_k = a_k/M$
    **if** $\alpha_k > \alpha_{\text{target}}$ **then**
      $\eta \leftarrow \eta \times f_{\text{increase}}$
    **else if** $\alpha_k < \alpha_{\text{target}}$ **then**
      $\eta \leftarrow \eta \times f_{\text{decrease}}$
    **end if**
  **end for**

---

We adopt default parameters: $f_{\text{increase}} = 1.1$, $f_{\text{decrease}} = 0.9$, $\eta_0 = 0.01$, $K = 200$, $K_{\text{burn-in}} = 100$, and $\alpha_{\text{target}} = 0.574$ for three unconditional tasks. For conditional tasks of VAE, we give more iterations of local search: $K = 500$, $K_{\text{burn-in}} = 200$.

It is noteworthy that by adjusting the inverse temperature $\beta$ into $R^\beta$ during the computation of the Metropolis-Hastings acceptance ratio $r$, we can facilitate a greedier local search strategy aimed at exploring samples with lower energy (*i.e.*, higher density $p_{\text{target}}$). This approach proves advantageous for navigating high-dimensional and steep landscapes, which are typically challenging for locating low-energy samples. For unconditional tasks, we set $\beta = 1$.

In the context of the VAE task (Table 2), we utilize two GFlowNet loss functions: TB and VarGrad. For local search within TB, we set $\beta = 1$, while for VarGrad, we employ $\beta = 5$. As illustrated in Table 2, employing a local search with $\beta = 1$ fails to enhance the performance of the TB method. Conversely, a local search with $\beta = 5$ results in improvements at the $\log \hat{Z}^{\text{RW}}$ metric over the VarGrad + Expl. + LP, even though the performance of VarGrad + Expl. + LP surpasses that of TB substantially. This underscores the importance of selecting an appropriate $\beta$ value, which is critical for optimizing the exploration-exploitation balance depending on the target objectives.

## E.2 Ablation study for local search-guided GFlowNets

**Increasing capacity of buffer.** The capacity of the replay buffer influences the duration for which it retains past experiences, enabling it to replay these experiences to the policy. This mechanism helps in preventing mode collapse during training. Table E.1 demonstrates that enhancing the buffer's capacity leads to improved sampling quality. Furthermore, Figure 1 illustrates that increasing the buffer's capacity—thereby encouraging the model to recall past low-energy experiences—enhances its mode-seeking capability.

Table E.1: Comparison of the sampling quality of each sampler trained with varying replay buffer capacities in Manywell. Five independent runs have been conducted, with both the mean and standard deviation reported.

| Buffer Capacity ↓ Metric → | $\Delta \log Z$ | $\Delta \log Z^{\text{RW}}$ | $\mathcal{W}_2^2$ |
|---|---|---|---|
| 30,000 | $4.41_{\pm 0.10}$ | $2.73_{\pm 0.15}$ | $6.17_{\pm 0.02}$ |
| 60,000 | $4.06_{\pm 0.05}$ | $2.38_{\pm 0.38}$ | $6.14_{\pm 0.04}$ |
| 600,000 | $4.57_{\pm 2.13}$ | $0.19_{\pm 0.29}$ | $5.66_{\pm 0.05}$ |

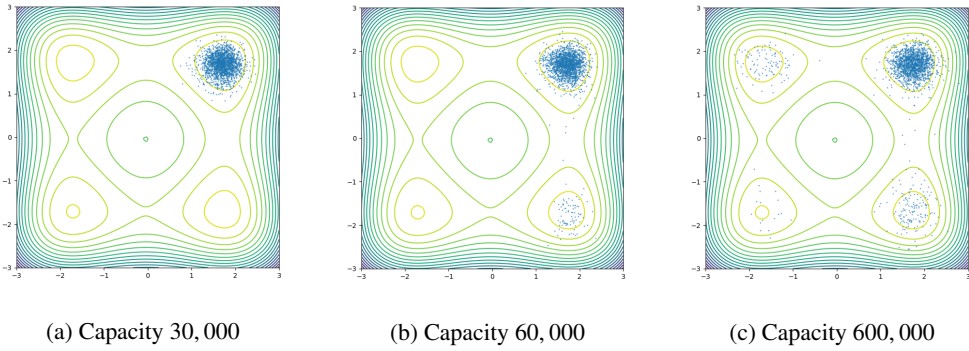

(a) Capacity 30,000      (b) Capacity 60,000      (c) Capacity 600,000

Figure E.1: Illustration of each sampler trained with varying capacities of replay buffers, depicting 2,000 samples. As the capacity of the buffer increases, the number of modes captured by the sampler also increases.

**Benefit of prioritization.**

Rank-prioritized sampling gives faster convergence compared with no prioritization (uniform sampling), as shown in Fig. E.2a.

**Dynamic adjustment of $\eta$ vs. fixed $\eta = 0.01$.** As shown in Fig. E.2b, dynamic adjustment to target acceptance rate $\alpha_{\text{target}} = 0.574$ gives better performances than fixed Langevin step size of $\eta$ showcasing the effectiveness of the dynamic adjustment.

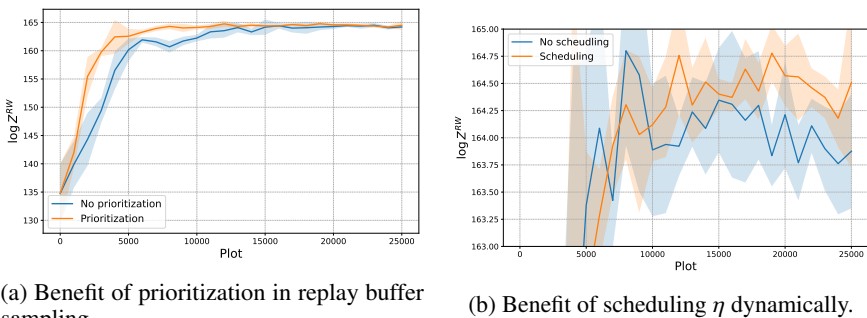

(a) Benefit of prioritization in replay buffer sampling..

(b) Benefit of scheduling $\eta$ dynamically.

Figure E.2: Ablation study for prioritized replay buffer and step size $\eta$ scheduling of local search. Mean and standard deviation are plotted based on five independent runs.

