# OpenReview forum: "Improved off-policy training of diffusion samplers"
_NeurIPS.cc/2024/Conference — NeurIPS 2024 poster_

### Official Review · Reviewer_FAJh · 2024-07-09

**Soundness:** 3
**Presentation:** 3
**Contribution:** 3
**Rating:** 7
**Confidence:** 5

**Summary:**

The paper studies the problem of training diffusion models to sample from a target distribution. The contributions are summarized as follows:

1. A codebase is provided for the study of diffusion-based samplers, due to the issue of inconsistent experimental settings in previous research;

2. Exploration in the target space can be enhanced by GFlowNet-based off-policy training objectives and local search with the use of replay buffer.

3. Experimental results validate the effectiveness of the proposed approach.

**Strengths:**

Sampling from a target distribution can be challenging in high-dimensional spaces, especially when the distribution of interest has many separated modes. This paper explores diffusion models to address this challenge. Unlike existing reverse KLD-based methods, such as PIS and DDS, this paper considers GFlowNet-based training objectives (e.g., trajectory balance, sub-trajectory balance), which enable off-policy training. This means that training trajectories are not necessarily from the current forward process, thus enhancing exploratory capability. Additionally, local search using a replay buffer can further enhance exploration in the target space. In general, the paper is well-written and well-organized.

---

**After rebuttal:** I will increase my score to 7. Typos or incorrect writing should be corrected upon acceptance.

---

**Weaknesses:**

Please see the below questions.

**Questions:**

- In terms of Table 1, which proposed methods perform best according to $\log Z^{LB}$ and $\log Z^{RW}$? Both evaluation metrics reflect different aspects of performance. For example, $\log Z^{LB}$ should indicate mode collapse behaviour, with lower values suggesting more serious mode collapse (e.g., PIS+LP: 13.19 vs. TB+Expl: 4.01, as illustrated in Figure 1)? In contrast, lower $\log Z^{RW}$ values suggest a closer approximation to the ground-truth $\log Z$.

- In terms of C2 variance-preserving noising process,

  - As far as I undertand from the paper, GFlowNet uses PIS architectures, where the initial state is a point mass at 0. How did you design it under VP settings, where the marginal distribution $p_{t}^{ref}$ is an invariant Gaussian distribution, i.e., the initial state is Gaussian distributed, not a point mass at 0?

  - In VP settings, we gradually add more and more noises to data (data --> Gaussian: $\beta$ increases). However, the sampling process starts with simple Gaussians and ends with samples, so should $\beta$ decrease, when Gaussian --> data?

- I am curious if the authors have ever tried the detailed balance objective, due to its superior generalization capability (https://arxiv.org/pdf/2407.03105, different settings though)?

- The results for DIS and DDS over Manywell are still missing compared to the previous version. However, the DIS paper does include the results. Any reasons?

- Missing reference - Beyond ELBOs: A Large-Scale Evaluation of Variational Methods for Sampling. Both papers share the same goal. It would be better to discuss it in the paper.


Potential typos:

- Eq.7: To define the reverse KL, a distribution is missing between $\int$ and $\log$? Why does $\mu_{0}(x_{0})$ appear before $d x_{\Delta t}$?

- Eq.8: $d x_{0}$ is missing?

- Eq.10 & 11: $p_{target}(x_{1})$ --> $R(x_{1})$?

- Line 181: $Z_{\theta}$ is missing in the equation?

- Eq.12: $f(x_{m})$ --> $f(x_{m \Delta t})$, as well as $f(x_{n})$?

- Line 209: $I_{d}$ is missing in $p_{t}(x_{t})$?

- Line 290 & 294: $\log$ is missing before $\int$, and $P_{F}(\tau | x_{1})$ --> $P_{F}(\tau | x_{0})$?

**Limitations:**

Yes. Limitations were included.

---

> ### Author Rebuttal · Authors · 2024-08-06
>
> Thank you for your comments and positive assessment of the paper.
>
> ### Evaluation metrics
>
> Thank you for your question regarding the different aspects of the methods' behaviours, as measured by various evaluation metrics (here: $\log \hat Z^{\rm RW}$ and $\log \hat Z$). We agree with your suggestions about the importance of one of them in measuring mode collapse behaviour and the other with the partition function approximation.
>
> However, for measuring both mode collapse and mode coverege, we have also a few other options. If we have access to ground truth samples, sample-based metrics (such as those typically used for assessing generative models trained from data) are available. One of them is $\mathcal{W}_2^2$, which we already included in Appendix C.1.
>
> Other options are, e.g., EUBO, ESS, MMD, and EMC, which are presented in the very interesting "Beyond ELBOs" paper you mentioned (which appeared after the submission deadline, but which we were already aware of). The question of which approach is the best in each metric is already hard. However, we noticed that $\log\hat Z^{\rm RW}$ results are better for the methods using Langevin Parametrization (e.g., together with Local Search), whereas $\log \hat Z$ performance is better when the method utilizes Local Search.
>
> ### VP setting
>
> Good question and observations! As you noticed, we used the PIS model architectures, but changed the noising (backward) process to add increasing levels of noise and made the initial state Gaussian-distributed and not a point mass at 0 (which can be thought of as a fixed first transition from an abstract initial state to a sample from the Gaussian). The exact procedure is presented in Appendix C.2.
>
> ### DB objective
>
> The DB objective fails to discover multiple modes even on the simplest task (25GMM) and thus performs significantly worse than SubTB, which is in turn worse than TB. We guess that this is due to poor credit assignment over long trajectories, as the learning signal to the policy is mediated by learning of the intermediate flow functions (densities).
>
> The preprint you mention tests generalization abilities only in a single environment, which (1) is discrete, (2) has much shorter and variable-length trajectories, (3) is simple enough that all learning algorithms converge to the optimum when modes are not masked. We guess that all three of these limit the generalizability of those results to the diffusion case.
>
> However, it would indeed be interesting, in future work:
> - to study the regularizing effect of local-in-time objectives like DB in the diffusion sampler setting, as has been done in the standard diffusion setting in [Lai et al., "FP-Diffusion", ICML'23];
> - to understand the meaning of these objectives in the continuous-time limit.
>
> ### Missing results
>
> We will include the missing results for DIS and DDS on Manywell. The main reason is that our (and, e.g., PIS's) Manywell setting is slightly different than the one used in the DIS paper. We are currently working on making DDS and DIS work on our Manywell version and plan to share those results before the end of the discussion period.
>
> ### Missing reference
> Thank you for pointing out this paper. We are aware of it and will happily include it in the revised version of our paper. (Note that that paper appeared only on June 11, after the NeurIPS submission date.)
>
> ### Potential typos
>
> Thank you for reading carefully! We fixed the small mistakes.
>
> **Thanks you again, and please let us know if we can provide any further information during the discussion period.**

---

> > ### Author Response · Authors · 2024-08-12
> > **Additional DDS+DIS results**
> >
> > As promised, here are the results for DDS and DIS on Manywell.
> >
> > First, we clarify the difference between the task as considered in DIS and in this work, noting that both have been studied in other papers as well and neither choice seems obviously better.
> >
> > |    | Ours | DIS (hardest version) |
> > |---|---|---|
> > | Dimension | 32 | 50 |
> > | No. of double wells | 16 | 5 |
> > | No. of modes | $2^{16}$ | $2^5$ |
> >
> > We consider Manywell as a product of 16 double well distributions having unnormalized density given by $R(x,y) = \exp(-x^4 + 6x^2 + 0.5x - 0.5y^2)$, following [Midgley et al., "Flow annealed importance sampling bootstrap", ICLR'23], whereas in the DIS paper, the authors consider the more general case of the distribution being a product of some number of double well distributions. However, despite their considering this problem in higher dimensionality, they are stacking a smaller number of double wells (no more than 5), which makes the number of modes in distribution much smaller.
> >
> > We ran DDS and DIS with our setting (after searching for the best hyperparameters, DDS was still somewhat unstable and often encounters an explosion of the loss and mode collapse, in which case we take results from the preceding checkpoint):
> >
> > | Algorithm | $\Delta\log Z$ | $\Delta\log Z^{\rm RW}$ | ${\cal W}_2^2$ |
> > |---|---|---|---|
> > | DIS (4 runs) | 10.52 ± 1.02 | 3.05 ± 0.46 | 5.98 ± 0.46 |
> > | DDS (3 runs) | 7.36 ± 2.43 | 0.23 ± 0.05 | 5.71 ± 0.16 |
> > | VarGrad + LP + LS (ours) | 4.11 ± 0.45 | 0.02 ± 0.21 | 5.30 ± 0.02 |
> >
> > We see that our best method outperforms both DIS and DDS in both metrics, but both are comparable with the various baselines in Table 1 of the paper.
> >
> > We hope you find this comparison helpful. Thank you again for your time.

---

> ### Comment · Reviewer_FAJh · 2024-08-12
>
> Thank you for your detailed responses, and I apologize for the delayed reply.
>
> ### Evaluation metrics
>
> - I agree that if ground-truth samples are available, then we can use sample-based metrics for evaluation.
>
> - I am still a bit confused. For example, if we look at TB + Expl. + LS (ours), we see that $\Delta \log Z = 0.171$ and $\Delta \log Z^{\mathrm{RW}} = 0.004$, which are the best in the column. We might conclude that it performs the best? It gives the lowest approximation error for $\log Z$, while suffering from more serious mode collapse?
>
> ### VP setting
>
> > Good question and observations! As you noticed, we used the PIS model architectures, but changed the noising (backward) process to add increasing levels of noise and made the initial state Gaussian-distributed and not a point mass at 0 (which can be thought of as a fixed first transition from an abstract initial state to a sample from the Gaussian). The exact procedure is presented in Appendix C.2.
>
> - Do you mean that the policy at the first step, from $s_{0} = ((0, 0), 0)$ to (x_{1}, 1) is constrained to be a unit Gaussian, i.e., $P_{F}((x_{1}, 1) | ((0, 0), 0)) := P(x_{1})$, while subsequent steps are conditional Gaussians with a known variance, i.e., $P_{F}(\cdot | (x_{t}, t))$?
>
> > In VP settings, we gradually add more and more noises to data (data --> Gaussian: $\beta$ increases). However, the sampling process starts with simple Gaussians and ends with samples, so should $\beta$ decrease, when Gaussian --> data?
>
> - In terms of the equation below Eq.(16), we see that $\beta$ increases from $0.01$ to $4$, implying that we are adding more and more noise for the forward policy. This approach works for diffusion generative modeling. However, the sampling process starts with simple Gaussians and ends with samples, so shouldn't $\beta$ decrease (i.e., $\beta$ decreases from $4$ to $0.01$)? This is my previous concern.
>
> [Updated] Thank you for providing new results.

---

> > ### Author Response · Authors · 2024-08-12
> >
> > Thank you for following up.
> >
> > **On the evaluation metrics:** we are not sure to understand your question. In fact, "TB + Expl. + LS (ours)" shows **less** serious mode collapse and correspondingly lower $\log Z$ estimation error.
> >
> > **On the VP setting:**
> >
> > > Do you mean that the policy at the first step ... is constrained to be a unit Gaussian?
> >
> > Yes, precisely as you wrote (assuming integer indexing of time steps).
> >
> > > However, the sampling process starts with simple Gaussians and ends with samples, so shouldn't $\beta$ decrease?
> >
> > We have mistakenly written the noise schedule in equation (16) **in reverse** (i.e., the time indexing convention for diffusion models trained from data, not the one for diffusion samplers). In fact, $\beta$ decreases linearly from 4 to 0.01 (as we have just confirmed in the code that was run for the experiments).
> >
> > Thank you again for reading so carefully.

---

> > > ### Comment · Reviewer_FAJh · 2024-08-12
> > >
> > > **On the evaluation metrics:** for example, TB+Expl: 4.01 vs. TB + Expl. + LS (ours): 4.57, together with Figure 1, where you mentioned that our proposed replay buffer with local search is capable of preventing mode collapse. I might conclude that TB+Expl suffers from more serious mode collapse, with giving a lower value?
> > >
> > > Other points are clear to me now. Thank you.

---

> ### Author Response · Authors · 2024-08-12
>
> That task (25GMM) is easy enough that none of the methods in the last group (using either LS or LP) are exhibiting mode collapse -- they sample points around each of the 25 modes.
>
> Note that:
> - a sampler that finds 9 of 25 modes (the inner 3-by-3 square) will have a $\log Z^{\rm RW}$ estimation error of about $-\log\frac{9}{25}\approx0.44$ (as TB and VarGrad without any extensions do);
> - a sampler that finds only one mode will have an error of about $-\log\frac{1}{25}\approx1.40$, and two modes $-\log\frac{2}{25}\approx1.10$ (and the methods with error >1 do indeed usually find one or two modes).

---

> > ### Comment · Reviewer_FAJh · 2024-08-12
> >
> > Thank you for taking the time to clarify. I am willing to increase my score to 7. Typos or incorrect writing should be corrected upon acceptance.

---

> > > ### Author Response · Authors · 2024-08-12
> > >
> > > Much appreciated, and thank you yet again for your detailed reading and finding all the small mistakes that we missed.

---

### Official Review · Reviewer_fepi · 2024-07-12

**Soundness:** 2
**Presentation:** 2
**Contribution:** 1
**Rating:** 3
**Confidence:** 3

**Summary:**

This paper proposes an off-policy diffusion-based sampler training method to match a target distribution and a corresponding exploration strategy and credit assignment to improve it.

**Strengths:**

1.	The proposed idea of this paper is interesting, which connects the Euler-Maruyama sampler and GFlowNets.

**Weaknesses:**

1.	Although the authors mention that traditional MCMC have high cost in sampling, the proposed method based on neural sde seems to still have this problem. To the reviewer’s knowledge, the solving procedure of neural sde is time-consuming as well.
2.	The experimental target distribution also seems relatively simple. In the reviewer’s opinion, for GMM, we can first sample a mode according to the weights of different modes and then obtain a sample in this mode. Hence, it seems unnecessary to use complex model like diffusion.
3.	In many real-world applications like image generation, the pdf (may be unnormalized) of the target distribution is unavailable and we can only achieve data samples from the target distribution. Hence, the application scenarios of the proposed model are limited.
4.	Besides, as mentioned in the conditional sampling case, the proposed method seems to need an extra trained vae to perform sampling. However, the vae can directly do the image generation. In that case, what is the real contribution of the proposed method?

**Questions:**

1.	The biological sequence design seems more appropriate to be the validation benchmark for the proposed model, which is also considered in GFlowNets. So the reviewer wonders how the proposed method is compared with GFlowNets in such tasks.
2.	Could the authors explain why they use the log-partition function estimation error as the metrics rather than the log-partition function itself? Similar to MLE (Maximum Likelihood Estimation), the model can be considered better with higher $\log Z$.

**Limitations:**

See Weakness 3.

---

> ### Author Rebuttal · Authors · 2024-08-06
>
> Thank you for your comments. Below we've tried to address what we believe to a number of misunderstandings.
>
> > Strength: The proposed idea of this paper is interesting, which connects the Euler-Maruyama sampler and GFlowNets
>
> First, we'd like to point out that this paper is not the first to connect Euler-Maruyama samplers of SDEs with GFlowNets (this was already done in [41] and a few other papers). The goal of this paper is to study methods that, for the first time, achieve results competitive with simulation-based methods on several continuous sampling benchmarks.
>
> Next, we answer the rest of the points:
>
> ### Applicability of diffusion samplers is limited
>
> We respectfully disagree:
>
> You are correct that there are many domains where we want to train a generative model from ground truth samples. That is an easier problem than sampling unnormalized densities, which is perhaps one reason for the greater attention it receives.
>
> However, many important Bayesian inference problems require sampling distributions that are only available through unnormalized density functions, and sampling methods for such distributions are an active area of research. As noted by the other reviewers:
> > The studied problem of sampling from a distribution is an important issue with a long history in statistical inference (**6eYp**)
>
> > Enabling diffusion models to be efficiently applied to sampling from unnormalized probability distributions is a problem with high potential for impact (**5SPJ**).
>
> A few such problems **beyond** Bayesian deep learning are simulations in statistical physics (e.g., [Nicoli et al., "Asymptotically unbiased estimation of physical observables with neural samplers", Physical Review E, 2020]), molecular dynamics problems (e.g., [Noé et al., "Boltzmann generators: Sampling equilibrium states of many-body systems with deep learning", Science, 2019]), imaging inverse problems in many scientific fields, etc. In fact, it is these problems that have motivated the benchmarks  that are adopted in the literature for **entirely data-free** inference methods such as ours (e.g., Manywell is a simple statistical physics model).
>
> ### Neural SDE integration is slow compared to MCMC
>
> This is inaccurate. The solution of a neural SDE requires as many neural net evaluations as the number of discretization steps chosen (for us, this is 100, in line with past work). This is *far* less expensive than the time to run MCMC chains to convergence on any nontrivial density.
>
> ### Target distributions are simple
>
> We kindly direct you to the response to all reviewers.
>
> ### Why test on a Gaussian mixture?
>
> Of course, you are correct that a Gaussian mixture density does not require a diffusion model to sample. However, 2D Gaussian mixture densities have been used as a benchmark in many past works ([41,85,87], among others) to test the ability to discover separated modes in an easily interpretable setting.
>
> ### The proposed method seems to need an extra trained vae to perform sampling
>
> This question appears to rest on a misunderstanding about what that experiment is showing. The goal there is to *learn an encoder for a trained VAE*, which is an established problem in the diffusion samplers literature (e.g., [87]). The goal is **not** to unconditionally sample new images, which could indeed be done by simply decoding noise samples with the pretrained decoder. Here, we need a pretrained decoder to provide the reward for the diffusion encoder that we train (the reward is the joint likelihood -- the product of the decoder likelihood and the prior density).
>
> ### Comparing with GFlowNets for biological sequence design
>
> There seems to be a misunderstanding here. Biological sequence design is a **discrete** sampling problem in which GFlowNets have indeed been used successfully. The goal of this paper is precisely to study the adaptation of GFlowNets and related methods to **continuous** sampling problems.
>
> ### $\log Z$ estimation error
>
> There may be some misunderstanding about the meaning of this metric. The metrics are **variational lower bounds** on $\log Z$, meaning that they are lower than the true log-partition function in expectation. The estimation error is a more meaningful metric than the raw value because the best possible value (0) is known and invariant to additive changes to the target log-density, which do not change the target distribution.
>
> When the true $\log Z$ is not known, one can report the estimate, not the estimation error, as we indeed do in some of the experiments (LGCP in Table 1 and VAE in Table 2).
>
> **Thank you again for your feedback. Please let us know if we have misunderstood anything in your comments and we will be happy to provide further clarification. If we have sufficiently clarified these points and answered your questions, please consider updating the score.**

---

> ### Comment · Reviewer_fepi · 2024-08-09
>
> Thanks for the detailed responses. However, the reviewer still considers it is necessary to do some experiments on high-dimensional real-world applications to show the applicability of the proposed method or make enough theoretical contribution like DDS anyway. Otherwise, it is hard to judge whether the proposed method is useful and efficient.
>
> In that case, the reviewer decide to reject this paper and wish the authors could find appropriate complex real-world applications for their method later. Moreover, I guess a potential application for your method is the robot locomotion task, which is considered in CFlowNets [1].
>
> [1] Li Y, Luo S, Wang H, et al. Cflownets: Continuous control with generative flow networks[J]. arXiv preprint arXiv:2303.02430, 2023.

---

> > ### Author Response · Authors · 2024-08-09
> >
> > Thank you for following up.
> >
> > As pointed out in [41, Section 3.1], the referenced paper [1] makes some serious errors in math -- incorrectly doing a change of variables in an integral -- that make the algorithm unsound. Formulating this task as a continuous sampling problem requires access to the Jacobian of the environment’s transition function, which isn’t available.
> >
> > As noted in the initial response, we have demonstrated the effectiveness of our sampling method on a standard collection of tasks in the diffusion samplers literature. If you see a task of higher difficulty that is standard in the field, please let us know.

---

### Official Review · Reviewer_6eYp · 2024-07-13

**Soundness:** 2
**Presentation:** 3
**Contribution:** 3
**Rating:** 4
**Confidence:** 2

**Summary:**

This paper focuses on the problem of sampling with distributions defined by a black-box and unnormalized energy function. This work provides a comprehensive review of existing works, including both variational methods and policy-based methods, and offers a codebase and benchmark to replicate and evaluate the existing works. Additionally, this work proposes a method to improve existing policy-based methods via local search and a replay buffer.

**Strengths:**

1. The studied problem of sampling from a distribution is an important issue with a long history in statistical inference. The paper provides a good review of recent works on this topic by leveraging diffusion models. The codebase that unifies existing methods is certainly useful to the community for continuing research on this topic.

2. The experiments are comprehensive in baselines, including not just diffusion-based methods but also classical MCMC algorithms. The results clearly show the advantages of diffusion-based methods and the techniques proposed in this work.

**Weaknesses:**

1. It appears to me that this work only tests the algorithm on relatively simple and manually-constructed scenarios. Are there any real and important applications within the field? I am not very familiar with this field, but I think that only conducting experiments on synthetic datasets makes this topic less practical. I believe the main advantage of the diffusion-based method over classical methods is in modeling complex distributions, making experiments on synthetic examples less meaningful.

2. Additionally, the tested scenarios are all low-dimensional cases. I wonder how this algorithm performs on high-dimensional cases, such as when the energy function is learned through neural networks. For example, is it possible to apply this algorithm to image generation where the energy function is represented by an image classifier? Testing the algorithm on high-dimensional tasks like these would provide a better understanding of its scalability and practicality in more complex and realistic settings.

**Questions:**

Is there any benchmark in this field involving a real application and complex high-dimensional distribution?

**Limitations:**

See my comments on the weakness above.

---

> ### Author Rebuttal · Authors · 2024-08-06
>
> Thank you for your comments, in particular, for acknowledging the strength of our comprehensive benchmarking.
>
> Regarding your questions about the benchmarks and their dimensionality, first, we kindly direct you to the response to all reviewers for discussion of the choice of target densities.
>
> Second, the dimensions of the tasks studied are: 2, 10, 20 (with 784-dim conditioning vector), 32, and 1600. As just noted, this is in line with past work on diffusion samplers. But it is also worth noting that sampling from complex densities in high dimension **without any prior knowledge** (such as a pretrained model or known samples from the target) is generally an unsolved problem, which is perhaps why it has not been the subject of much study in this past work. Our methods, which use off-policy training, can be combined with prior knowledge, such as known target samples (such as by placing those samples in the replay buffer), as has already been demonstrated, e.g., in [42]. Simulation-based methods like PIS, DDS, etc. do not have this ability.
>
> The use of diffusion samplers in settings with prior knowledge, while not the subject of this paper, is indeed a very interesting direction of research and likely important in more difficult applications.
>
> **Thank you again for your review. Please let us know if we can provide any other clarification that would help you in understanding and assessing the paper.**

---

> > ### Author Response · Authors · 2024-08-13
> >
> > Dear Reviewer 6eYp,
> >
> > This message is to ask if you have any final questions and if our response has affected your assessment of the paper.
> >
> > Thank you,
> >
> > The authors.

---

### Official Review · Reviewer_5SPJ · 2024-07-17

**Soundness:** 3
**Presentation:** 3
**Contribution:** 2
**Rating:** 6
**Confidence:** 3

**Summary:**

The paper presents a variety of improvements to off-policy strategies for training diffusion models to sample from unnormalized densities. Equation 13. These include maintaining a replay buffer (obtained with Langevin sampling) to enable efficient off-policy exploration and incorporating an inductive bias into the neural network which estimates the SDE drift term. They also present a software library containing unified implementation of these techniques and e.g. diffusion model training.

**Strengths:**

- Enabling diffusion models to be efficiently applied to sampling from unnormalized probability distributions is a problem with high potential for impact
- Thorough experimental analysis and comparison of different alterations to the training procedure. On most problems considered, the authors' contributions are necessary to achieve good results with trajectory balance.
- The contribution of a software library could be valuable to the community.

**Weaknesses:**

- The results are not overwhelming - although the proposed contributions are helpful compared to a basic version of TB, there is only one modeling task in Tables 1-2 (25GMM) where they provide a statistically significant improvement over the baselines.
- The experiments are on synthetic energy functions and MNIST VAE. Including more real-world data or models would be informative.

**Questions:**

See weaknesses. My main concern is the underwhelming results when compared to DIS, DDS, PIS, PIS+LP. Is there any reason why the proposed method(s) should be preferred to these? Or e.g. expected to scale better to more complex problems?

**Limitations:**

Adequately addressed

---

> ### Author Rebuttal · Authors · 2024-08-06
>
> Thank you for your review. Below we will answer your questions and concerns.
>
> ### Statistically significant improvement over the baselines
>
> Firstly, we want to highlight that our method achieved comparable or better results to current SOTA models. In Tables 1 and 2, we highlighted the best results (red) and those that are not statistically-significantly worse according to the Welch test (blue). Whereas the baselines achieve good results usually in one or two tasks, our method and its variations are the best or among the best results in **nearly all tasks simultanously**.
>
> ### Experiments on synthetic energy functions
>
>
> We thank the reviewer for their suggestion on including the real-world sampling example. Note that the LGCP task is already a real-world example (it is derived from observations of pines in Finland). However, we recognize the importance of applications and will add more experiments (e.g., molecular conformations) in the revision.
>
> We kindly direct you to the response to all reviewers for further discussion of the choice of densities.
>
> ### Underwhelming results when compared to SOTA?
>
> Simulation-based methods, such as PIS, are predominantly on-policy, making it challenging to explore low-energy regions in high-dimensional spaces and prone to mode collapse in complex densities (see Figure 1 in the Manywell task). In contrast, training a sampler using off-policy techniques allows for the utilization of powerful methods like local search, which enhances exploration and performance. This off-policy trick has been verified in various problems, including simple energy (25GMM), complex multimodal energies (Manywell), high-dimensional tasks (LGCP), and conditional posterior sampling tasks (VAE).
>
> **Thanks again for your feedback. Please let us know if you have any more questions and if any further clarifications would help you assess the paper.**

---

> > ### Author Response · Authors · 2024-08-13
> >
> > Dear Reviewer 5SPJ,
> >
> > This message is to ask if you have any final questions and if our response has affected your assessment of the paper.
> >
> > Thank you,
> >
> > The authors.

---

> > ### Comment · Reviewer_5SPJ · 2024-08-13
> >
> > Thank you for the rebuttal. After considering it, I believe that this paper does present a valuable contribution through its thorough comparisons and the generally-good performance of the proposed method. I still believe that it would be strengthened by experiments on more complex/high-dimensional problems but given that, as the author's point out in the rebuttal, the evaluations tasks are standard in the continuous sampling community, I believe they are sufficient for acceptance. I have raised my score to 6.

---

### Author Rebuttal · Authors · 2024-08-06

We thank all the reviewers for the effort they put into reviewing our paper and are grateful for the constructive feedback. We appreciate the reviewers remarking that the paper is well-written and well-organized (FAJh), studies a important problem (5SPJ, 6eYp), and does comprehensive benchmarking to demonstrate the effectiveness of the algorithms (6eYp).

Thank you for noting that the paper well-written and well-organized (FAJh), studies a important problem (5SPJ, 6eYp), and does comprehensive benchmarking (6eYp).

Several reviewers (5SPJ, 6eYp, fepi) raised concerns about the difficulty/dimensionality of the densities studied. We respond to those points now.

First, we note that **our evaluation followed the common procedure in the continuous sampling community** [8,41,77,85,87,etc.], which typically evaluates on the same tasks that we chose, with some variations, or even on a subset of them. For instance, DDS and DIS papers evaluate the methods also only on synthetic data and VAE/NICE (DDS) or sampling one image (DIS).

Furthermore, **as a secondary contribution, we identified discrepancies and reproducibility issues in past work** on these benchmarks (see Appendix B.1). We believe that this is important for sound progress in this field in the future.

The dimensions of the tasks studied are: 2, 10, 20 (with 784-dim conditioning vector), 32, and 1600, which is indeed less than the dimension on which diffusion models are typically trained from data, e.g., images. However, sampling from densities in high dimension **without any prior knowledge** (such as a pretrained model or known samples from the target) is a harder problem than maximizing [a variational bound on] the likelihood of samples, which is perhaps why it has not been the subject of much study in this past work.

Our methods, which use off-policy training, can be combined with prior knowledge, such as known target samples (such as by placing those samples in the replay buffer), as has already been demonstrated, e.g., in [42]. Simulation-based methods like PIS, DDS, etc. do not have this ability. The use of diffusion samplers in settings with prior knowledge, while not the subject of this paper, is indeed a very interesting direction of research and likely important in more difficult applications.

---

### Decision · Program_Chairs · 2024-09-25

**Decision:**

Accept (poster)

**Comment:**

This paper studies diffusion-structured samplers for amortized inference over continuous variables. The authors present a detailed survey of existing methods for diffusion-structured sampling, and draw connections between several methods in the literature, placing them in a common framework.

The paper makes two main contributions: 1) it provides a software library unifying existing methods and performs a comprehensive benchmark of different diffusion-structured samplers, investigating unconditional and conditional generative modeling tasks: sampling from various energy distributions with dimensionalities ranging from 2 to 1600, and conditional sampling from the latent posterior of a VAE; and 2) they propose a new exploration strategy for off-policy methods that uses local search and a replay buffer.

The authors show that their proposed exploration method helps prevent mode collapse and achieves competitive or better performance than other methods on several unconditional and conditional modeling tasks. Their benchmark results also identify reproducibility issues with previous work, which is useful for the field.

Reviewers found that the paper is well-written, addresses an interesting and important problem, and provides comprehensive benchmark results. Several noted that the provided software library unifies existing methods and will be valuable to the community.

A few reviewers mentioned that the problems in the experiments are small-scale; however, the authors addressed this, noting that their evaluation setting is in line with previous work.

Overall, this paper provides valuable contributions that will be of interest to the NeurIPS community.